# Impacts on tundra vegetation from heavy metal-enriched fugitive dust on National Park Service lands along the Red Dog Mine haul road, Alaska

Peter N. Neitlich[1]*, Shanti Berryman[1¤a], Linda H. Geiser[2], Anaka Mines[1¤b], Alyssa E. Shiel[3]

1 National Park Service, Alaska Regional Office, Anchorage, Alaska, United States of America, 2 United States Department of Agriculture-United States/Forest Service, Washington, District of Columbia, United States of America, 3 College of Earth, Ocean, and Atmospheric Sciences, Oregon State University, Corvallis, Oregon, United States of America

¤a Current address: Independent Researcher, Santa Fe, New Mexico, United States of America
¤b Current address: Independent Researcher, Twisp, Washington, United States of America
* peter_neitlich@nps.gov

**Data Availability Statement:** Data are publicly available from the National Park Service data repository: https://irma.nps.gov/DataStore/

## Abstract

The DeLong Mountain Transportation System (DMTS) haul road links the Red Dog Mine—one of the world's largest zinc mines—with a shipping port on the Chukchi Sea in northwest Alaska, USA. The road traverses 32 km of National Park Service (NPS) lands managed by Cape Krusenstern National Monument (CAKR). Fugitive dusts from ore concentrate transport and mining operations have dispersed zinc (Zn), lead (Pb), cadmium (Cd), and metal sulfides onto NPS lands since the mine began operating in 1989. This study assessed the effects of metal-enriched road dusts on the diversity and community structure of lichens, bryophytes, and vascular plants in dwarf-shrub tundra within CAKR. In a Bayesian posterior predictions model, lichen species richness (LSR) was highly correlated to distance from the haul road and was distributed on the landscape consistently with the spatial patterns of Zn, Pb and Cd patterns published earlier in this journal. The mean modeled LSR of the 3000–4000 m distance class was 41.3, and LSR decreased progressively down to 9.4 species in the 0–50 m class. An ordination of 93 lichen species by 91 plots revealed strong community patterns based on distance from the haul road. The major community gradient was highly correlated ($r = 0.99$) with LSR and negatively correlated with Cd, Pb and Zn ($-0.79 < r < -0.74$). Ordinations of bryophyte classes showed less response than lichens to distance from the road and heavy metals values, and vascular plant ordination showed less still. Measures of bryophyte health such as the midrib blackening and frond width of *Hylocomium splendens* were positively correlated with distance from the haul road and negatively correlated with this same suite of elements. A total area of approximately 55 km² showed moderate to strong impacts on lichens from fugitive dusts. This is equivalent to an area of almost 1 km on both sides of the haul road running 32 km through CAKR.

Reference/Profile/2293058. Companion data found
in Neitlich et al. 2017 are available at: https://irma.
nps.gov/DataStore/Reference/Profile/2240112.

**Funding:** This study was funded by the National
Park Service Arctic Network (Inventory and
Monitoring Program) and the Western Arctic
National Parklands. The funders had no role in
study design, data collection and analysis, decision
to publish, or preparation of the manuscript.

**Competing interests:** The authors have declared
that no competing interests exist.

## Introduction

Lichens represent a large portion of the biomass and diversity of Arctic Alaska's tundra ecosystems [1–3]. Together the lichens and bryophytes of the northwest Alaskan national parks contribute a species diversity greater than that of vascular plants, and a co-dominance in cover [4]. The sensitivity of lichens to air pollution and to heavy metals has been well-established for decades [5], and for this reason lichens have been widely used as pollution biomonitors [6, 7]. Due to their dependence on atmospheric sources of nutrients and water, both lichens and mosses readily uptake atmospheric inputs [8]. This has made both groups valuable as passive monitors of nitrogen (N), sulfur (S), metals, and semi-volatile organic compounds [7, 9]. While knowledge of damage to lichens, bryophytes and vascular plants from metal smelting is well documented in the Arctic [10, 11], less is known about the effects of fugitive dusts from mining operations on vegetation. This study examines the impacts of escaped mine dusts containing elevated levels of zinc (Zn), lead (Pb) and cadmium (Cd) on tundra ecosystems in Cape Krusenstern National Monument (CAKR), a U.S. National Park Service (NPS) unit in northwest Alaska.

CAKR encompasses approximately 2,660 km$^2$ of coastal plain and montane habitats along the Chukchi Sea in northwest Alaska (Fig 1). This conservation unit is located in the continuous permafrost zone [12] and hosts a variety of arctic tundra vegetation types from wet lowlands to alpine barrens. In the cold and harsh environments of NPS's Arctic parks, lichens and bryophytes represent approximately 70% of the species of the tundra plant communities and sizeable fractions of the plant cover [4, 13, 14]. The Red Dog Mine, one of the largest Zn and Pb mines in the world, is located approximately 50 km northeast of the monument boundary and 10 km from the boundary of Noatak National Preserve (also administered by the NPS). The mine has operated year-round since 1989 to produce Zn and Pb concentrates (approximately 50–55% concentration) in fine powder form. Concentrates are hauled on a continuous basis year-round in 80 ton covered trucks from the Red Dog Mine Site to the concentrate storage buildings at the Red Dog Port Site via the Delong Mountain Transportation System (DMTS) haul road (also known as the Red Dog Mine haul road and hereafter referred to as simply the "haul road"). The concentrates are then transferred to barges for shipping during the ice-free shipping period each year, usually July through October.

The 84 km haul road traverses approximately 32 km of CAKR land located within an industrial use easement designated by Congress [15] in 1985 in an amendment to the monument's enabling legislation [16] and overseen by the Alaska Industrial Development and Export Authority (AIDEA). Concentrate haul trucks and other vehicular traffic had been dispersing fugitive dusts onto NPS lands for approximately 17 years at the time of this field work [17–19]. These dusts are enriched with, Zn, Pb, Cd, and S, all in reduced sulfide forms [20]. Metal-laden muds and dusts originate in the mine pit, waste rock areas, and concentrate loading and unloading facilities [20]. Once attached to vehicles, they become dispersed onto the roadbed and surrounding tundra on the trips between the mine and port sites [21]. Mud sampled from the wheel wells of a personnel transport vehicle—which would not routinely enter ore concentrate handling areas and would, therefore, be expected to have less contamination than concentrate haul trucks—contained between 18,000–23,000 mg/kg Zn and 10,000 mg/kg Pb [21]. The roadbed surface contained approximately 500–600 mg/kg Zn and 100 mg/kg Pb in 2006 [21]. Soils near the port site (a few kilometers from NPS lands) contained levels of Zn, Pb and Cd as high as 15,000, 10,400 and 71 mg/kg respectively at the time of this research [20], which led to zones of dead and/or stressed vegetation adjacent to the Concentrate Storage Buildings.

From 1991 to 2000, concentrate haul trucks were covered only with tarps. In 2001, a new fleet of trucks was purchased with sealed lids, and in 2003 procedural modifications were

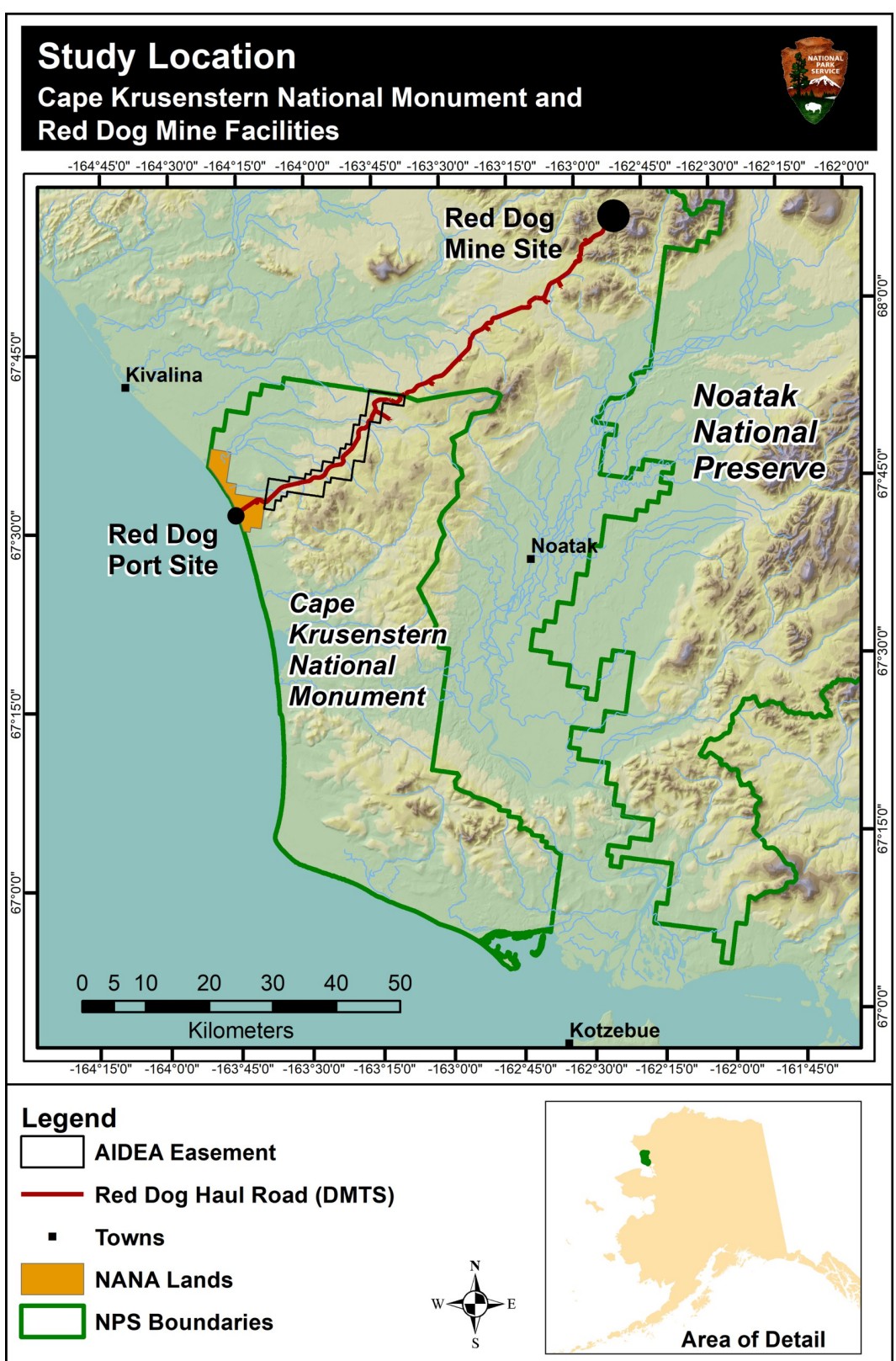

**Fig 1. Study area: Delong Mountain Transportation System haul road and Cape Krusenstern National Monument, Alaska.** Thirty-two km of the haul road traverses an industrial easement through the monument. Geographic coordinates are

N for latitude and W for negative longitude. Congress granted the Northwest Alaska Native Association (NANA) an easement through the monument in 1985. NANA retained ownership of lands including the Port Site located within the monument boundary.

incorporated at the port facility to reduce concentrate dust losses during unloading and loading operations [20]. Truck washing was also installed for the three to four months of the year with temperatures above freezing, which helped reduce dispersal of fugitive dusts during those months.

CAKR was established as an NPS unit in 1980 [16] in part "to protect habitat for and populations of, birds, and other wildlife, and fish resources; and to protect the viability of subsistence resources." NPS's Organic Act [22] and subsequent NPS Management Policies [23] requires of all NPS units that "the National Park Service will preserve and protect the natural resources, processes, systems, and values of units of the national park system in an unimpaired condition to perpetuate their inherent integrity and to provide present and future generations with the opportunity to enjoy them." The 99 year industrial easement through CAKR granted to the Northwest Alaska Native Association (NANA) detailed four clauses which would be relevant to the fugitive dust issue. First, the National Park Service would retain ownership of easement lands, but the lands in the easement could be used by NANA (and its proxy, AIDEA) "as if the lands had been conveyed to NANA." Second, the easement required NANA to protect fish, wildlife and their habitat during mining operations. Third, the easement required the recipient to "use all reasonable means" to protect both air and water quality and to comply with Alaska Department of Environmental Conservation (ADEC) standards for these resources. Fourth, the easement owner was required to take dust control measures, in consultation with NPS, to protect air quality as required by ADEC. In sum, while the NANA easement was expected to have some level of impact from mining operations, Congress also mandated substantial environmental protections on easement lands. Lands in CAKR but outside of the industrial easement were expected to be protected as originally designated in the monument's enabling legislation.

The moss *Hylocomium splendens* (Hedw.) Schimp. has been used for several decades as a passive sampler to document patterns of airborne heavy metal deposition in northern areas [24] and on NPS lands from Red Dog mining operations in particular. In 2000, elevated concentrations of Cd ($>10$ mg/kg) and Pb ($> 400$ mg/kg) were found in *H. splendens* along the haul road corridor [19]. The following year, strong gradients in Zn, Pb, and Cd deposition were reported in CAKR and found to be associated with the haul road, the Port Site and the Mine Site [18]. Heavy metal levels in moss were highest immediately adjacent to the haul road (Zn$>1500$ mg/kg, Pb $> 900$ mg/kg, Cd $> 24$ mg/kg). Finally, in 2006, research associated with the current project reported decreases of up to 50% in deposition of Zn, Pb and Cd between 2001 and 2006 in areas close to the road [17]. Decreases appeared to follow operational controls including dust mitigation on roads and infrastructure, hydraulically-covered trucks, and seasonal truck washes. However, heavy metals deposition is expected to be ongoing for the life of the mine and possibly beyond.

In 2007, consultants for the mine published an extensive ecological risk assessment (ERA) documenting potential risk factors to different ecosystem compartments from heavy metal-enriched fugitive dust [20]. The ERA concluded that out of a large array of potential faunal receptors, only small mammals and Willow Ptarmigan (*Lagopus lagopus)* showed increased risks of mortality due to exposure to toxins. Vegetation receptors, by contrast, were strongly affected by fugitive dusts. Nonvascular vegetation cover was thought to be reduced out to 2 km

from the haul road compared to reference site cover, though no detailed analysis of lichen or bryophyte community structure was conducted.

While dozens of lichenological studies have demonstrated injury and mortality of lichens in the presence of heavy metals, sulfate and sulfides from both industrial point sources (e.g., smelters, mines) and regional plumes [11, 25], there has been little research on the effects of fugitive dusts enriched with heavy metals on lichens or bryophytes. In Alaska, the only studies of dust impacts to lichens pertain to crustal dust rich in Al, Fe, and Ca on the Dalton Highway (i.e., the Prudhoe Bay haul road, [26, 27]. This study showed strong effects on lichens, but the effect distance was limited to within approximately 50 m from the road.

The current study is a part of our broader 2006 field work that led to our earlier publication on spatial patterns of heavy metals deposition in CAKR [17]. Our goals for this study were: 1) to examine the relationship between lichen, bryophyte and vascular plant community structure (e.g., species richness, cover) vs. levels of Zn, Pb and Cd deposition (which are closely related to the distance from the haul road), 2) to explore the differences in response to fugitive dust contaminants among the different vegetation components (i.e., lichens, bryophytes, vascular plants), 3) to model lichen species richness (LSR) spatially along the haul road, 4) to map LSR and determine an effect area in terms of LSR quantiles, and 5) to explore the relationship of nonvascular plant vigor with other covariates. Monitoring of plant community condition and contaminants in CAKR is part of the National Park Service Arctic Network's long term monitoring protocols [28], and this study is intended to be repeated decadally to assist NPS with its conservation mandates.

## Methods

### Ethics statement

This study was approved by the National Park Service-Western Arctic National Parklands interdisciplinary project review team under the provisions of the National Environmental Policy Act [29]. The project was assessed to be low in impacts to natural resources, cultural resources and subsistence and was granted a categorical exclusion from further review.

### Study area

This study of plant community structure occurred in conjunction with (and at the same sites as) the field work for the remeasurement of spatial patterns of heavy metal deposition in 2006 [17]. The study area (Fig 2), contained within 4000 m of the haul road within CAKR, is described in detail in that research article. In brief, CAKR is located along the Chukchi Sea on a coastal plain tundra ecosystem dominated by an open, low, dwarf shrub-cottongrass (*Eriophorum* L.) tussock tundra [14] interspersed with well-drained hills supporting a variety of alpine lichen, forb, and dwarf shrub species. The mean annual temperature is approximately −7˚C [30]. Summer temperatures in the region typically fluctuate between 2˚ and 18˚C, while temperatures along the Chukchi Sea coast range from 4˚ to 13˚C [20]. Winter temperatures typically range from −26˚ to −15˚C. The modeled mean annual precipitation along the haul road in CAKR ranges from 320 to 380 mm, with montane areas receiving up to 510 mm [31]. Soils are poorly developed due to the cold climate, low precipitation, and the near-continuous permafrost [14]. Where exposed, bedrock is predominantly calcareous. Mean monthly winds at Kotzebue, about 100 km south of the port site, are above 10 knots from September through April and blow from the east [32]. Mean wind speeds are comparable during the summer months (with an average of 10.5 knots) but are from the west.

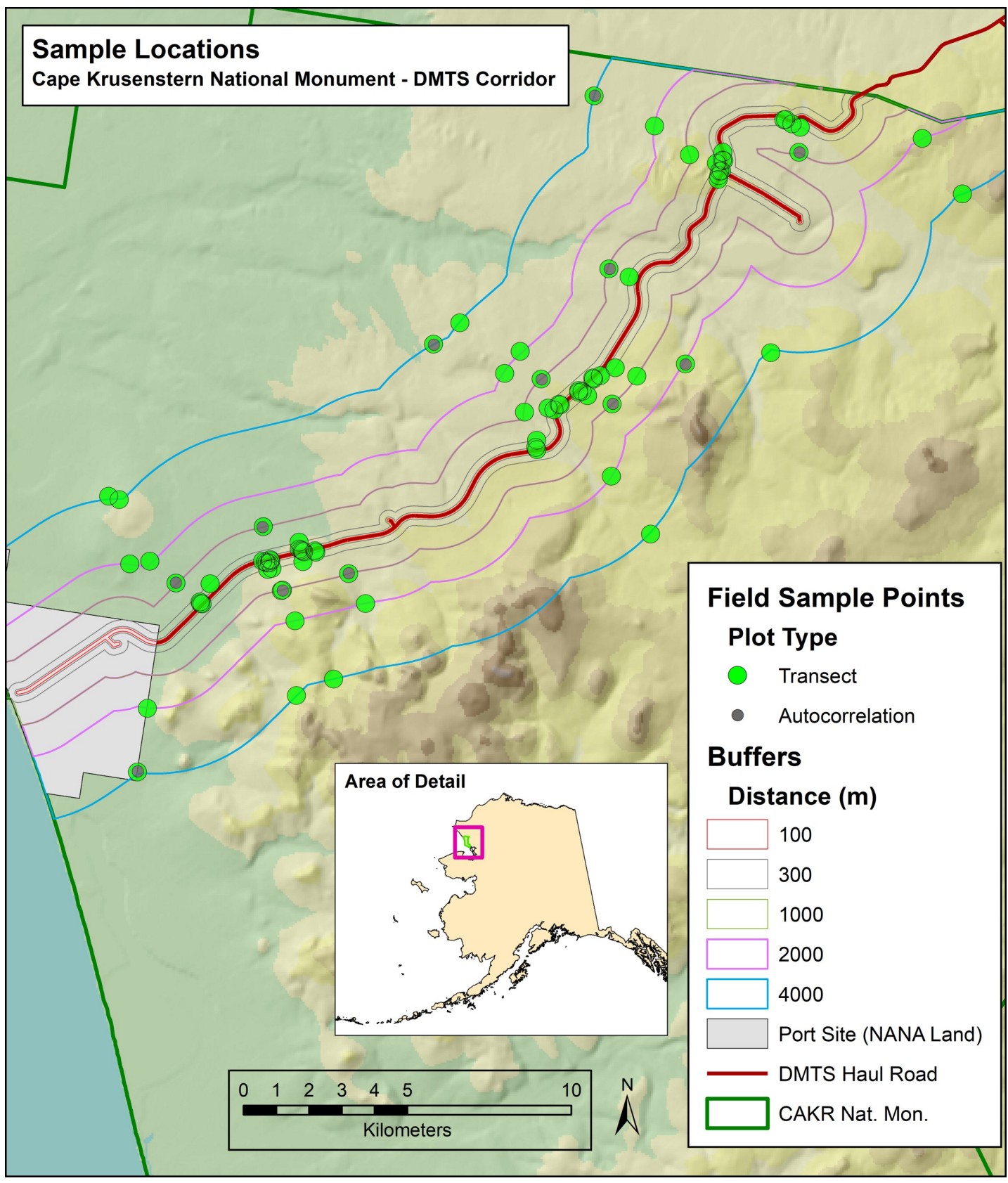

**Fig 2. Sample plot locations in Cape Krusenstern National Monument, Alaska.** Plots were arrayed at distances of 10, 50, 100, 300, 1000, 2000, and 4000 m from the DMTS haul road using GIS-based haul road buffers at those distances. Autocorrelation plots were 10–20 m from selected transect plots in the 1000–4000 m distance classes.

## Study design

Our primary goal was to use species-based multivariate community analysis (e.g., ordination, indicator species analysis, multivariate tests of community differences) to relate changes in vegetation community structure (e.g., species presence, abundance, diversity, cover) to covariates including contaminant levels, environmental variables, and distance from the haul road. Hasselbach et al. [18] found that deposition of heavy metals decreased logarithmically as a function of distance away from the haul road. We established permanent monitoring plots based on distance classes from the road that would reflect this log-based contaminant decay function (Fig 2), sampling from 10 to 4,000 m out from both sides of the haul road.

The study design controlled for natural differences in vegetation communities and elevation, using a stratified-random design based on the two dominant land cover types within a moderate elevation range. Vegetation along the haul road includes lowland wet tussock tundras, sedge-dominated wetlands, upland dwarf shrub tundras, willows, and, in a few locations, alpine dwarf shrub communities. The most common land cover types classified by Jorgenson et al. [14] are Upland Moist Dwarf Birch-Ericaceous Shrub and Upland Moist Dwarf Birch-Tussock Shrub. Together, these closely related types accounted for 66% of the area along the haul road out to the farthest sampling distance of 4000 m and these types were fairly rich in lichens and bryophytes—which had shown high sensitivity to Zn, Pb, S, and alumino-silicate road dust in other studies [e.g., 5, 27]. These two cover types are quite similar except that the latter has greater than 15% cottongrass (*Eriophorum vaginatum* L.) cover while the former is co-dominated by dwarf birch (*Betula nana* L.) and ericaceous shrubs. To maximize the signal from contaminant effects and to minimize the noise from natural variation in community structure across different types of vegetation, we limited our sampling to these two dominant types. Vegetation communities in CAKR change considerably with elevation [14], with lichens generally increasing in cover and species richness with increasing elevation. The haul road stays at a fairly low elevation throughout CAKR and never directly traverses any of the nearby alpine communities. To avoid confounding the pollution signal with natural variation in vegetation due to elevation, we excluded alpine communities and restricted our sampling to between 60–250 m in elevation ($\bar{x}$ = 130 m). Mean elevations of plots were not significantly different between distance classes or transects in ANOVA.

Sampling distances from the road were established in an attempt to obtain an even spread of values along the heavy metals deposition gradient in the posterior prediction models reported by Hasselbach et al. [18]. Twelve "transects" were created according to the distances in Table 1 of Neitlich et al. [17], with points at the following distances (in m): 10, 50, 100, 300, 1000, 2000, and 4000 (Fig 2). One autocorrelation plot was established in each transect at a random distance 10 to 20 m from plots in the 1000, 2000 or 4000m distance classes in each transect to characterize local spatial variability. Each "transect" consisted of a group of points at specified distance from the road and from within one of the two dominant land cover types chosen in ArcGIS 9.2 [33] along a straightest possible line.

Two groups of reference plots were chosen at random from points in CAKR with sufficient representation of the two desired land cover types, and at a distance of at least 20 km from the haul road. Each reference area had 3 replicates of each land cover type, making for a total of 12 pre-chosen reference plots. Of these, 10 of these were ultimately surveyed (Table 1 in Neitlich et al. [17]). Upon visitation, it became evident that most of the reference plots had been

**Table 1. Mean values of important co-variates at the full range of distance classes from the DMTS haul road in CAKR, with the standard error of the mean (presented in parentheses).** Mean elemental values are mg/kg in *Hylocomium splendens* tissue, dry weight, except % Total S and % Total N. Values are presented for the Reference Sites and the Kotzebue road sites that were not included in subsequent analyses of vegetation community structure. The 2001 values for Cd, Pb and Zn are from Hasselbach et al. [18].

| | Distance Class | | | | | | | | |
|---|---|---|---|---|---|---|---|---|---|
| | 10 | 50 | 100 | 300 | 1000 | 2000 | 4000 | Ref | Kotz |
| N | 12 | 12 | 12 | 12 | 20 | 12 | 14 | 10 | 3 |
| Mean Cd, 2001 (modeled) | 8.7 (1.1) | 7.7 (1.0) | 7.0 (0.9) | 4.2 (0.5) | 1.4 (0.1) | 1.0 (0.1) | 0.6 (0.0) | 0.1 (0.0) | - |
| Mean Cd | 5.4 (0.5) | 3.5 (0.4) | 2.7 (0.3) | 1.4 (0.1) | 0.7 (0.1) | 0.6 (0.1) | 0.6 (0.1) | 0.2 (0.0) | 0.4 (0.1) |
| Mean Pb, 2001(modeled) | 437 (45) | 358 (35) | 311 (33) | 170 (17) | 39 (4) | 30 (5) | 18 (2) | 1.8 (0.1) | - |
| Mean Pb | 179 (21) | 99 (15) | 67 (10) | 28 (4) | 13 (1) | 12 (2) | 10 (1) | 1.7 (0.5) | 2.1 (0.4) |
| Mean Zn, 2001(modeled) | 1410 (138) | 1217 (122) | 1073 (111) | 633 (66) | 191 (17) | 151 (20) | 100 (9) | 41 (0.5) | - |
| Mean Zn | 566 (68) | 331 (44) | 239 (32) | 113 (11) | 75 (7) | 77 (7) | 77 (9) | 46 (5) | 32 (5) |
| Mean Al | 9,070 (526) | 4,896 (721) | 3,026 (430) | 1,144 (224) | 425 (58) | 346 (52) | 204 (14) | 253 (41) | 1,253 (497) |
| Mean Ca | 28,319 (3,104) | 16,908 (2,282) | 11,498 (1,699) | 5,927 (661) | 4,199 (409) | 4,184 (365) | 3,555 (180) | 4,802 (387) | 6,716 (2,024) |
| Mean Fe | 15,463 (1155) | 7,841 (1037) | 4,641 (537) | 2,044 (342) | 724 (81) | 598 (76) | 367 (20) | 355 (49) | 3,150 (1365) |
| Mean % S | 0.16 (0.01) | 0.10 (0.01) | 0.08 (0.00) | 0.07 (0.00) | 0.06 (0.00) | 0.06 (0.00) | 0.06 (0.00) | 0.07 (0.00) | 0.06 (0.00) |
| Mean %N | 0.62 (0.02) | 0.77 (0.02) | 0.84 (0.03) | 0.84 (0.02) | 0.94 (0.02) | 0.95 (0.03) | 0.96 (0.01) | 0.99 (0.04) | 0.82 (0.04) |
| Mean slope (deg) | 1.6 (0.2) | 1.9 (0.2) | 2.2 (0.3) | 2.1 (0.4) | 2.1 (0.3) | 3.6 (0.7) | 3.9 (0.8) | 6.4 (1.7) | 0.3 (0.3) |
| Mean LSR | 4.8 (2.0) | 13.8 (2.2) | 18.1 (2.0) | 25.2 (2.1) | 32.1 (1.9) | 37.3 (1.7) | 36.3 (2.7) | 28.2 (3.0) | 34.3 (2.2) |
| Mean cover bryophytes (%) | 7.5 (1.4) | 13.4 (1.7) | 15.0 (2.6) | 16.2 (2.6) | 13.8 (1.4) | 10.2 (1.5) | 7.4 (0.9) | 6.0 (0.9) | 7.4 (2.2) |
| Mean cover lichens (%) | 0.6 (0.2) | 2.7 (0.7) | 2.7 (0.6) | 6.7 (1.4) | 9.3 (1.5) | 5.3 (0.5) | 8.5 (1.4) | 10.0 (1.6) | 10.7 (1.7) |
| Mean % of Hyl spl midrib blackening | 72 (8) | 24 (4) | 10 (2) | 13 (5) | 2.1 (0.7) | 1.1 (0.5) | 0.9 (0.4) | 0.1 (0.0) | 14.5 (7.5) |
| Mean width Hyl spl (cm) | 1.3 (0.0) | 1.5 (0.0) | 1.4 (0.1) | 1.4 (0.0) | 1.5 (0.0) | 1.5 (0.1) | 1.5 (0.0) | 1.6 (0.0) | 1.5 (0.2) |
| Distance to Road Centerline | 17.6 (0.3) | 57.7 (0.2) | 103.8 (1.9) | 296.7 (3.1) | 999.1 (5.3) | 2019 (10.9) | 4000 (11.3) | 56241 (2550) | 100413 (0.7) |

misclassified in the land cover map [14]. While they are useful for examining background metals levels (e.g. [17]) and nonvascular plant vigor traits, they were not useful for vegetation comparison and they were thus omitted from the study. Lastly, 3 plots were installed along a dusty, gravel road (Ted Stevens Way) outside of Kotzebue, AK, approximately 110 km from the haul road for exploratory purposes, but again, were not included in the primary vegetation analysis.

## Field methods

Sample plot locations were determined in GIS and uploaded into Trimble GeoExplorer XH and GeoExplorer XT GPS units [34]. Researchers located each point on the ground with a real-time estimated horizontal position error ranging from 10 to 200 cm and permanently marked and recorded each location using GPS. Prior to plot establishment, locations for plots 10 and 50 m from the haul road were adjusted with a measuring tape from the roadbed's toe slope. This added approximately eight additional meters of distance from the road centerline to the points in the 10 m and 50 m distance classes, and was done to insulate these plots from potential roadbed widening (potentially leading to big differences between plots in vehicle proximity at this close range, and high mud splatter or burial) and/or roadbed sloughing, thus avoiding potential issues for repeat monitoring. Due to moderate horizontal error of both real-time GPS and the GIS centerline feature, the final placement of these closer plots by tape also what otherwise could have been high proportional error within these distance classes.

Permanent plots were surveyed as follows: A corner stake was marked with a permanent rebar spike in one corner, and a 4 x 8 m rectangular plot was then established with the long axis parallel to the haul road [28]. One hundred points arrayed on a grid of 40 x 80 cm were surveyed for vegetation in a point-intercept method using an underwater laser pointer on a staff. At each point, each plant species or other substrate (e.g., mineral soil, duff, water, litter) touched by the laser beam was recorded as 1% cover. If multiple strata of live plants occurred along the laser beam, each was recorded as 1%. After the 100 points were completed, the plot was additionally surveyed for taxa not detected in the point counts. Each taxon detected only during this "walkaround" was assigned 0.1% cover. Macrolichens and vascular plants were either assigned a species name in the field or vouchered (off plot if possible) for later identification. Bryophytes were not recorded to species, but rather assigned to five morphological groups: *Sphagnum* spp., *Hylocomium splendens*, acrocarpous mosses (i.e., those with upright stems with reproductive structures at the tips), pleurocarpous mosses (i.e., those with decumbent, branching and/or feather-like growth forms and reproductive structures borne at the tips of lateral branches) other than *Hylocomium*, and liverworts. These groups are easily distinguished in the field by staff without extensive bryological training.

Slope and aspect were recorded for the plot. Height measurements were taken for 5 replicates (where available) of the following lichen species: *Cetraria cucullata*, *Cetraria laevigata*, *Cladina arbuscula/mitis*, *Cladina stygia*, *Thamnolia subuliformis/vermicularis*. Height measurements were made by carefully removing a wet specimen from the ground without fragmenting it and measuring from the tip to the basal decay zone. Whole plot ocular cover estimates were made for the following variables: tall shrubs, dwarf shrubs, forbs, sedges, *Carex/Eriophorum*, grasses, bryophytes, lichens, bare soil, duff/organic matter, rock, and standing water. These variables were also recorded by the point counts if they occurred on a point.

Within 10 m of each plot (but at the same distance to the haul road), a collection of approximately 2 l of the moss *Hylocomium splendens* was made for contaminants analysis in the moss tissues. Field duplicates (i.e., nearly identical moss collections from same location and microsites) were collected in 10% of the sample sites. Sample preparation and analysis was reported in Neitlich et al. [17], which described the spatial patterns of heavy metals deposition obtained from these samples.

## Laboratory analysis

The moss tissue samples were analyzed for Al, B, Ca, Cd, Cr, Cu, Fe, Mn, Mg, Ni, P, Pb, Zn, Cl, $NO_3^-$, $SO_4^{+2}$ total N, and total S at the University of Minnesota Research Analytical Laboratory following the protocols described in S1 File of Neitlich et al. [17]. Analysis included laboratory duplicates, standard reference materials, and lab QA/QC evaluation. Chlorophyll a, Chlorophyll b, and phycocyanin were determined by the U.S. Geological Survey-Columbia Environmental Research Center using spectrophotometry ASTM Standard Method 10200 H [35]. $SO_4^{+2}$ and $Cl^-$ were analyzed using USEPA 9056a [36].

## Taxonomy

Taxonomy followed the 7th North American Lichen Species Checklist [37]. Species concepts followed Goward [38] and McCune and Geiser [9] for fruticose lichens and Goward et al. [39] and McCune and Geiser [9] for foliose lichens. Vascular plant taxonomy and species concepts followed Hultén [40].

Because of the difficulty of correctly assigning cover for certain closely related lichen chemo-species or isomorphic species, several species were grouped in the field. *Thamnolia vermicularis and T. subuliformis* almost always co-occur in arctic vegetation plots [1, 41] and were

grouped under *T. subuliformis*. *Cladonia arbuscula* and *C. arbuscula* ssp. *mitis* are similarly co-occuring and impossible to separate in the field. These were grouped as *C. arbuscula*. *Dactylina arctica* and *D. beringica* were lumped under *D. arctica*. *Cladonia borealis* was lumped with *Cladonia coccifera* as only thin-layer chromatography (TLC) reliably differentiates these. There is considerable taxonomic difficulty separating *Cladonia maxima* from *Cladonia gracilis* ssp. *vulnerata* in the field. These were grouped under the name *Cladonia maxima sensu lato*. *Cladonia squamosa* var. *subsquamosa* was lumped with *Cladonia squamosa*. Separations between *Cladonia bacilliformis* and *C. cyanipes* were made primarily on size: following Goward, specimens up to 25 mm were called *C. bacilliformis*, and >25mm were called *C. cyanipes*. Anything obviously on wood was called *C. bacilliformis*. One highly unusual and widespread form of *Cladonia cornuta* with squamulose podetia was grouped with that species even though this morph is rare for *C. cornuta*.

There was a small number of specimens identifiable only to genus. These were retained for species richness analysis because those genera included many other species present in the region that were not otherwise present in the data set but presumed to be likely to occur. Specimens of known genus and unknown species—and for which all species in that genus occurred at the site—were lumped with the most common species in the data set. Examples of this included *Sphaerophorus* and *Bryoria*.

## Community analysis

**Lichens.** Community analysis via nonmetric multidimensional scaling ordination (NMS) was tested in a variety of configurations using PC-ORD 7 [42] to obtain best fit with the data (i.e., highest variance explained, lowest stress, highest orthogonality, lowest skew). Trials using "slow and thorough" autopilot settings without a pre-determined number of axes included use of untransformed, log transformed, or classed data, use of both Euclidean and Sorenson distance measures, and use of Bray-Curtis polar ordination as an alternative.

In the original matrix of 107 plots x 113 lichen species, there were three empty plots (i.e., without lichen species), all at 10 m, and several other plots at 10 m with only one or two species. The coefficient of variation was 87% for plots and 190% for species in the raw data, suggesting the need for eliminating rare species in the community analysis [43]. To examine patterns on the unmodified data set, outlier analysis was run with Euclidean distance measure, the only measure allowing input matrices with empty plots [42]. This test detected seven outliers greater than two standard deviations from the mean distance. The outlying plots were all highly diverse plots from distances > 300 m, predominantly 1000 to 4000 m and reference plots. Empty plots were deleted and outlier analysis was re-run using Sorenson distance. This time 6 plots > 2 standard deviations were identified, five from 10 m and one from 50 m.

The disparity in outlier analysis results suggested that a log or a square root transformation could help to rein in values at both the diverse and impacted ends of the gradient. Each cover value, x, in the plot by species matrix was log transformed to a new cover value ($x_{new}$) as follows:

$$x_{new} = \left(\log_{10}(x + 0.01)\right) + 2 \qquad (1)$$

This equation returned former zero values to zero after the transformation. Prior to ordination, the 3 plots with no species and 22 species with one or fewer occurrences were deleted from the species matrix. Reference plots, while mapped in classified imagery as belonging to the same land cover types as the transect plots [14], were in actuality wetter, more mid-height shrub-dominated, divergent in forb species and less diverse in lichens than the similarly-classed upland plots along the haul road. Multi-Response Permutation Procedures (MRPP,

[43]), a multivariate test of community differences showed substantial differences between the reference plots and the plots of 1000–4000 m from the haul road. (MRPP is computationally similar to ANOVA in that it compares matrix dissimilarity within and among groups.) Accordingly, we omitted the reference plots from the final ordination. A species-area curve with the new matrix of 91 plots x 93 species confirmed the adequacy of diversity capture with the plots present.

Out of three ordination trials using PC-ORD on autopilot mode with Sorenson distance measure using the "slow and thorough" setting, we chose the model with the best balance of high percent variance explained (88%), low final stress (15.9) and low final instability (0.007) high orthogonality (100%), and high correlations with covariates. This PC-ORD setting conducts 250 trials with real data and 6 starting dimensions and 250 trials with randomized data.

MRPP was run for lichen, vascular plant and bryophytes community matrices to quantify the strength of differences in community structure among different distance groups. NMS ordination was run on a vascular plant data set of 94 plots by 42 species after log transforming the main matrix in the same manner as the lichen community matrix and deleting plots and species with fewer than 2 occurrences to optimize fit. Orthogonalities ranged from 98.5% to 99% among axis pairs. Indicator species analysis [44] was run to determine which taxa were most associated with different distances from the road or contaminant levels.

In the analysis of the correlations of elemental concentrations with the ordination, tissue values for two plots from which no *Hylocomium* tissue could be found (T1N300 and E4N50, from 300 and 50 m, respectively) were modeled for use in PC-ORD, which prohibits blank values. The missing values were calculated as the mean of the 5 other values from the same side of the road at that distance. In instances where more than one lab sample was obtained for a plot (e.g., lab or field duplicates), the average value was used.

**Bryophytes.** As preliminary univariate analysis showed little correlation between the log of distance to road vs bryophyte cover ($r^2 = 0.01$) and because bryophytes were heavily grouped into dominant species and classes, we did not spatially model bryophyte abundance. There were, however, large differences in the community structure of bryophyte groups with respect to key covariates. Preliminary NMS trials showed weakly structured data suggesting the need to relativize by species maximum. A final $\log_{10}$-transformed bryophyte matrix of 94 plots by 5 species groups relativized by species group maximum yielded a two-dimensional solution that explained 87 percent of the variance in the data set. The final stress was 15.8, with a final instability of $< 0.000001$ and an orthogonality of 98%.

## Spatial analysis

A spatial model of lichen species richness was created using a geostatistical spatial model in a generalized linear model setting [45, 46]. A prediction grid described in Neitlich et al. [17] was used for posterior predictions. The spatial coordinate system was based on the Transversal Mercator projection using a central meridian that was the mean of all of the longitude coordinates of our data. Explanatory variables included logarithm of distance from haul road, side of road (i.e., north vs. south), and the two dominant land cover types. The response variable $Y_i$ was the count of number of different species in the $i$th plot, and it was modeled with a Poisson distribution, $Y_i \sim \text{Poi}(\lambda_i)$, where $\lambda_i$ is the Poisson mean parameter. The model is made spatial in the generalized linear model setting by the log link function and spatially autocorrelated errors,

$$\log(\lambda_i) = \beta_0 + \beta_1 d_i + \delta_i + \eta_i + \varepsilon(s_i) \tag{2}$$

where $\beta_0$ and $\beta_1$ are parameters, $d_i$ is log distance from road, $\delta_i$ is an effect for side of road

(north or south), $\eta_i$ is an effect for habitat (Ericaceous Dwarf Shrub Tussock or Dwarf Shrub Tussock Tundra), and $\varepsilon(s_i)$, is a random error at location with spatial coordinates $s_i$. The random errors are assumed to be spatially autocorrelated with

$$C(h) = \theta_i(h = 0) + \theta_2 \exp(-h/\theta_3) \qquad (3)$$

where $h$ is the distance between any two points, $I(a)$ is the indicator function (equal to 1 if the expression $a$ is true, otherwise it is 0), and the vector $\theta$ contains three parameters: the nugget $\theta_1$, partial sill $\theta_2$, and range $\theta_3$.

We used a Bayesian approach to inference, as adopted by Diggle et al. [46], using Markov chain Monte Carlo (MCMC) methods [47, 48]. We used very diffuse normal priors on the regression parameters $\beta_0, \beta_1, \delta_i$ and $\eta_i$, and the reference prior recommended by Berger et al. [49] for the covariance parameters. Convergence was assessed via visual checks of sample history plots and calculation of estimates of the potential scale reduction factor [50]. The maximum scale reduction factor for all parameters was 1.04, indicating good MCMC convergence, after a burn-in of 1000 iterations. After burn-in, the chains had 20,000 samples, and each 100[th] iteration was used as a sample (200 total samples) from the posterior distribution for each parameter.

In addition to using MCMC to fit the model, we computed predictions on the grid of points used in Neitlich et al. [17] for contaminant mapping. These same posterior predictions methods were used in Hasselbach et al. [18], but here we extend it to the generalized linear model setting. The 200 samples were used to compute mean predicted values, their confidence intervals, and functions of predictions such as the area above a threshold.

## Nonvascular plant vigor

The health and vigor of nonvascular plants was assessed with five measurements: chlorophyll *a* and phycocyanin in *Hylocomium*, percent of midrib blackened and frond width in *Hylocomium* collected for lab analysis, and heights of dominant lichens on plots. *Hylocomium* measurements were obtained on 10 randomly selected individuals within each plot's target collection of 2 l of that species. Percent blackening of the midrib of *Hylocomium* was assessed visually in increments of 5%. The dry 0frond width of *Hylocomium* was measured at its maximum extent. Lichen heights were obtained by measuring at least 5 randomly chosen replicates of a given species in full extension, but excluding any obviously dead material on sampled thalli.

## Results

### Lichen communities

**Lichen diversity and cover.** A total of 113 macrolichen taxa were found in the 94 plots surveyed (S1 Table). LSR ranged from 0 (three occurrences at 10 m) to 56 species (1 occurrence at 4000 m). In the raw (unmodeled) data, the mean LSR ranged ordinally from approximately 5 species per plot at 10 m from the haul road to 37 species per plot at 2000 m (Fig 3). Mean lichen cover ranged from 0.6% close to the road to between 5–9% for plots between 1000–4000 m from the road (Table 1). Mean lichen cover was intermediate (approximately 3%) at the 50 and 100 m distances. In non-spatial linear models lichen cover showed highly significant differences by distance classes (F = 7.7, p<0.0001). There were three distance class groups within which cover was statistically equal at p<0.05: 10 m, 50–100 m, and 300–4000 m (S2 Table). Linear regression of log lichen cover versus log distance to road had a correlation of $r^2 = 0.48$. Due to vegetation type mismatch, the reference plots were not useable for diversity

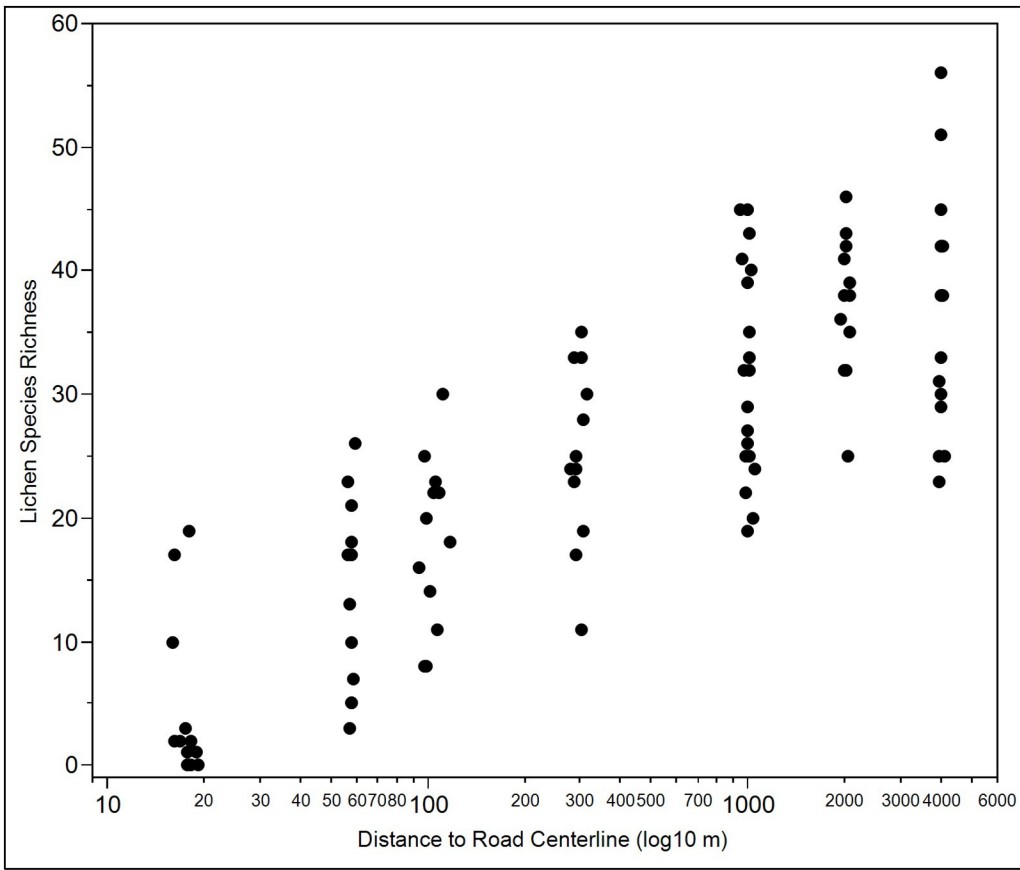

**Fig 3. Lichen species richness versus the $\log_{10}$ of distance to the DMTS haul road in Cape Krusenstern National Monument, Alaska.** Lichen species richness in 94 vegetation plots (4 x 8 m) between 10–4000 m from the road, Cape Krusenstern National Monument, Alaska.

comparisions, but were retained for other comparisons of environmental co-variates and metrics of nonvascular plant vigor below.

**Environmental covariates.** The concentrations in *Hylocomium splendens* of crustal elements (Al, Fe, Ca), heavy metals (Zn, Pb, Cd), and the abundance or response of numerous biological indicators changed markedly from the 10 m to the 4000 m distance class (Table 1). In the raw (unmodeled) data, Zn concentrations, for instance, ranged from a mean of 566 mg/kg at 10 m from the haul road to 77 mg/kg at 4000 m and 46 mg/kg at the Reference Sites. A detailed report of the spatial patterns of Zn, Pb, and Cd deposition modeled by posterior prediction are presented in Neitlich et al. [17]. In the modeled data, Zn ranged from 563 mg/kg in the 0–100 m stratum to 70 mg/kg in the 2000–4000 m stratum. Without consideration of spatial covariation, all of the elemental concentrations in Table 1 had highly significant differences between distance classes in ANOVA (p<0.0001), as did LSR, lichen cover, bryophyte cover, and the % blackening of *H. splendens*.

A correlation matrix (S3 Table) provides a sense of the strength of correlation between a suite of elements and a subset of key biological variables, without consideration of spatial relationships. As expected many elements were highly correlated and such variables as LSR, $\log_{10}$ lichen cover, and % of *Hylocomium splendens* midrib blackening were strongly correlated to a suite of elemental values. Both mean LSR and log lichen cover were strongly correlated with $\log_{10}$ distance to the haul road (r = 0.82 and 0.71, respectively). The percent of *H. splendens*

midrib blackening was negatively correlated with $\log_{10}$ distance to the road (r = -0.75). $\log_{10}$ elemental concentration of the heavy metals Cd, Pb, Zn, Cu, Ni, Mg and of Fe and Ca were highly correlated to each other generally. Pb and Zn were the most highly correlated elements in the data set both as modeled values in 2001 (r = 0.996) and as measured point values in 2006 (r = 0.99). Heavy metal and crustal element concentrations were highly correlated with $\log_{10}$ distance to road (-0.94 < r < -0.74), consistent with previous work establishing heavy metal gradients away from the road [17, 18]. LSR was slightly more highly correlated with the 2006 Zn point values than the 2001 Zn modeled values (r = -0.79 vs. -0.74).

**Community analysis.** Our chosen NMS ordination of 91 transect plots x 93 lichen species along the haul road explained 88% of the variance with 79% loaded on Axis 1 (Fig 4). The ordination was rotated to align Axes 1 and 2 maximally with the most highly correlated variables and to obtain 100% orthogonality of ordination axes. Since LogLSR (followed by LSR and many contaminant concentrations) was the variable most strongly correlated to Axis 1—and of strong interest—the ordination was re-rotated to align with LogLSR while leaving the perfect orthogonality intact.

Plots grouped in a fairly ordinal fashion according to distance class from the haul road, with the 10–100 m group and the 1000–4000 m group forming distinct, non-overlapping clusters. Axis 1 was strongly correlated with logLSR (r = 0.96) and lichen species richness (r = 0.85, Table 2) and strongly negatively correlated with the elements associated with fugitive dusts. Because community composition was relatively homogeneous other than the effect of many species disappearing close to the road, lichen species richness itself appears to have been the major driver for this axis. Elements having a strong negative association with Axis 1 were a

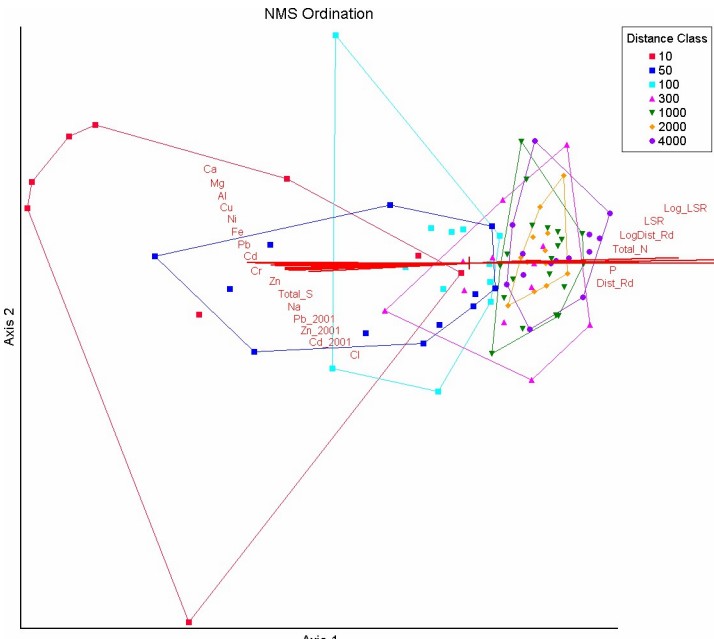

**Fig 4. Nonmetric multidimensional scaling ordination of lichen plots at distances 10 m through 4000 m from the DMTS haul road in Cape Krusenstern National Monument, Alaska.** The ordination is overlain by vectors proportional to the strength of their correlation with ordination axes. Spearman r values for each variable are provided in Table 2. Convex hulls surround plots in each distance class. All two letter abbreviations are elements. Mean elemental values in the data were mg/kg in *Hylocomium splendens* tissue, dry weight, except % Total S and % Total N. Other abbreviations are: Total_S (total sulfur), LSR (lichen species richness), LogDist_Rd ($\log_{10}$ distance to the haul road), Total_N (total nitrogen), P (phosphorus), Dist_Rd (distance to the haul road).

**Table 2. Spearman correlation coefficients (r) of environmental variables for two NMS ordinations on transect plots out to 4000 m from DMTS haul road in CAKR: Lichen ordination (91 plots x 93 species) and bryophyte ordination (91 plots x 5 species groups).** Values for which r > 0.5 are presented in bold.

| | LICHEN | | BRYOPHYTE | | | LICHEN | | BRYOPHYTE | |
|---|---|---|---|---|---|---|---|---|---|
| | Axis 1 | Axis 2 | Axis 1 | Axis 2 | | Axis 1 | Axis 2 | Axis 1 | Axis 2 |
| Variable | r | r | r | r | Variable | r | r | r | r |
| **VEGETATION** | | | | | **ELEMENTAL CONCENTRATIONS** | | | | |
| **Log Lichen Species richness** | **0.964** | 0.139 | **0.554** | 0.055 | **Al** | **-0.822** | -0.087 | **-0.651** | -0.015 |
| **Lichen species richness** | **0.845** | 0.139 | **0.515** | 0.02 | B | 0.211 | 0.248 | 0.114 | -0.095 |
| Lichen Cover | 0.316 | -0.077 | 0.004 | 0.116 | **Ca** | **-0.869** | 0.069 | **-0.627** | -0.059 |
| Vascular species richness | 0.263 | 0.221 | 0.018 | -0.305 | **Cd** | **-0.791** | -0.121 | **-0.594** | -0.009 |
| Tall Shrub Cover | -0.334 | 0.082 | -0.115 | -0.015 | **Cr** | **-0.793** | -0.05 | **-0.671** | -0.002 |
| Dwarf Shrub Cover | -0.099 | 0.069 | -0.189 | -0.086 | **Cu** | **-0.811** | -0.09 | **-0.591** | -0.001 |
| Forb Cover | 0.191 | 0.051 | -0.008 | 0.107 | **Fe** | **-0.806** | -0.098 | **-0.637** | -0.007 |
| Sedge Cover | -0.056 | -0.116 | 0.056 | 0.088 | K | -0.188 | 0.241 | -0.093 | -0.123 |
| Grass Cover | 0.058 | 0.134 | -0.064 | -0.049 | **Mg** | **-0.853** | 0.021 | **-0.587** | -0.044 |
| Bryophyte cover | 0.045 | 0.004 | 0.072 | -0.236 | Mn | 0.394 | -0.005 | **0.553** | 0.139 |
| Hylocomium Width | 0.238 | -0.03 | | | **Na** | **-0.701** | -0.037 | **-0.596** | -0.038 |
| **Hylocomium Midrib %Black** | **-0.752** | 0.016 | | | **Ni** | **-0.813** | -0.064 | **-0.654** | 0.009 |
| Chlorophyll a* | -0.305 | -0.129 | -0.206 | -0.124 | **P** | **0.532** | 0.088 | 0.432 | -0.033 |
| Phycocyanin* | -0.173 | -0.057 | -0.04 | -0.136 | **Pb** | **-0.792** | -0.138 | **-0.577** | -0.018 |
| | | | | | **Zn** | **-0.739** | -0.168 | **-0.563** | -0.013 |
| **ENVIRONMENT** | | | | | **Zn 2001** | **-0.65** | -0.068 | **-0.553** | -0.031 |
| **Log Distance to Road** | **0.761** | 0.083 | **0.652** | 0.071 | **Pb 2001** | **-0.682** | -0.063 | **-0.561** | -0.032 |
| Distance to Road | 0.493 | 0.1 | 0.499 | 0.104 | **Cd 2001** | **-0.585** | -0.069 | **-0.512** | 0.022 |
| Slope | 0.21 | 0.005 | 0.153 | -0.055 | **Total S** | **-0.717** | -0.094 | **-0.561** | 0.018 |
| Aspect | 0.014 | 0.094 | -0.072 | -0.052 | **Total N** | **0.676** | 0.118 | **0.574** | -0.002 |
| Longitude | -0.128 | -0.091 | -0.016 | 0.038 | Cl⁻ mg/l | -0.471 | 0.056 | -0.296 | -0.121 |
| Latitude | -0.174 | -0.112 | -0.046 | 0.041 | $NO_3^-$ mg/l | 0.033 | 0.208 | 0.171 | 0.107 |
| Elevation | -0.013 | -0.14 | 0.083 | 0.004 | P mg/l | 0.464 | 0.013 | 0.465 | 0.094 |
| Bare soil cover | -0.115 | 0.129 | -0.031 | 0.168 | $SO_4^{+2}$ mg/l | 0.173 | 0.204 | 0.087 | -0.077 |
| Duff / organic soil cover | -0.199 | 0.169 | -0.325 | -0.035 | | | | | |
| Standing water cover | -0.081 | 0.134 | 0.051 | 0.144 | | | | | |

whole suite of crustal and heavy metals associated with road dust and ore concentrates (–0.74 ≥ r ≥ –0.87). Zn had a strong negative association with Axis 1 (–0.739) as did Total S (–0.717). The modeled 2001 contaminant values had lower correlation coefficients than the measured contaminant values on plots in 2006. The log of distance to the road, closely tied to contaminant concentrations (per Neitlich et al., [17]), had a strong positive association with Axis 1 (r = 0.761). The lichens positively correlated with this axis and in which r > 0.5 were the terrestrial and epiphytic lichens: *Cladonia cornuta*, *Cetraria sepincola*, *Vulpicida pinastri*, *Cladonia fimbriata*, *Cetraria laevigata*, *Flavocetraria cucullata*, *Cladonia sulphurina*, *Cladonia uncialis*, *Cladonia amaurocraea*, and *Cladonia maxima* (S4 Table). The percentage of *Hylocomium splendens* midrib blackening was also strongly associated with Axis 1 (r = –0.762), and the blackening of this species in plots close to the road was visible to observers and in photos (Fig 5). While most lichens species were eliminated at distances close to the road (10–50 m), those present included: (> 50% frequency) *Thamnolia subuliformis*, *Flavocetraria cucullata*, *Cladonia amaurocraea*, and (33–50% frequency) *Cladonia stygia*, *Cladonia arbuscula*, *Cetraria laevigata*, *Cladonia maxima*, *Cladonia rangiferina*, *Peltigera aphthosa*. These are the species in the study most tolerant to pollutants potentially including Zn, Pb, Cd, and sulfides.

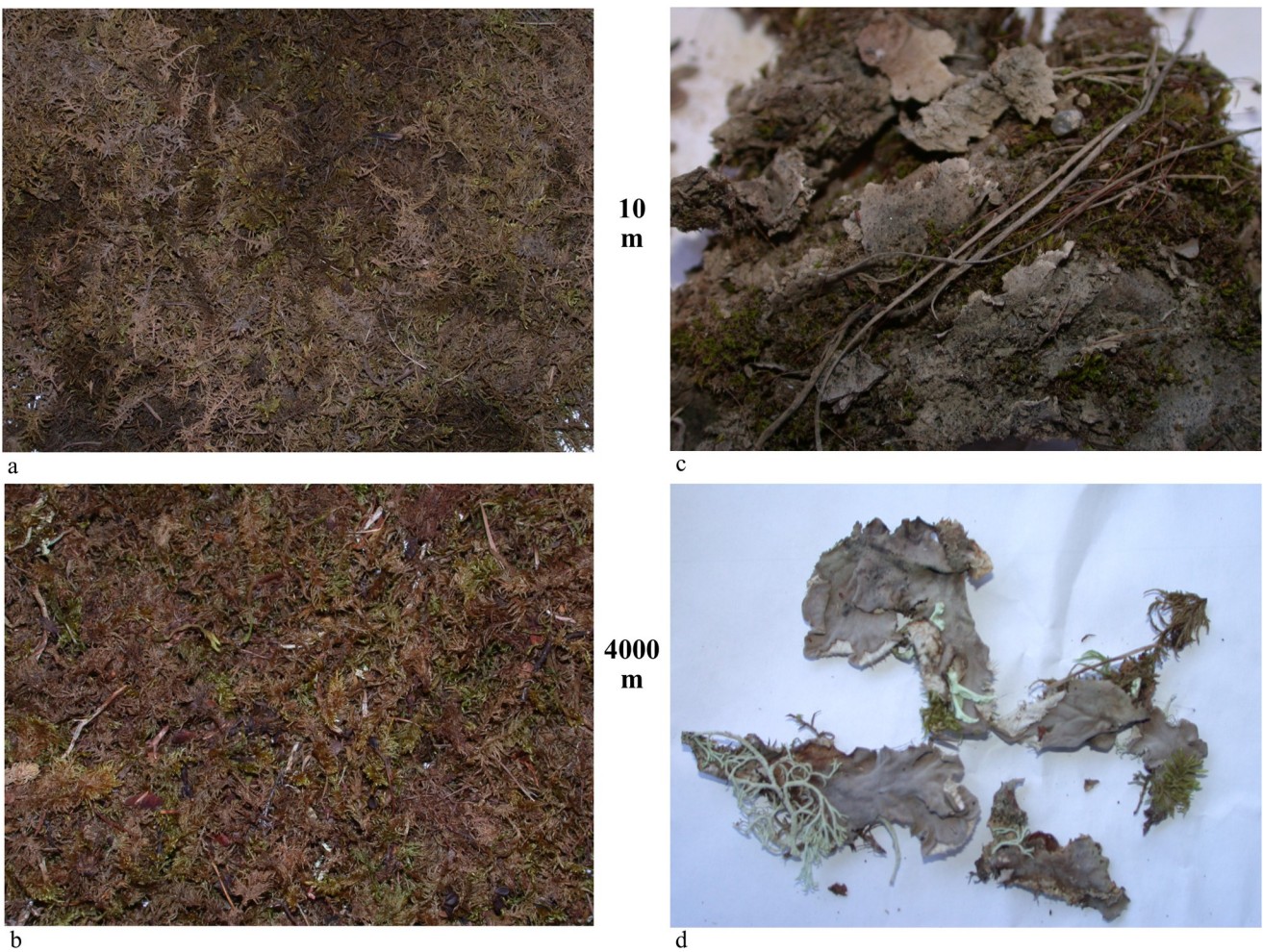

**Fig 5. Photos of *Hylocomium splendens* moss and *Peltigera* sp. lichen from 10 m vs 4000 m away from the DMTS haul road in Cape Krusenstern National Monument, Alaska.** Photos a, b: *H. splendens*. Photos c,d: *Peltigera* sp. Photos were taken in the lab using dried materials.

Aside from distance to the road and the elemental concentrations in *Hylocomium splendens*, none of the non-lichen environmental variables had strong associations with either axis, suggesting that these were not strong drivers of community patterns. Axis 2 had no strong associations with environmental variables or with any lichen species, suggesting that this axis had little interpretable value other than accounting for additional unexplained variance. The wide spread of plots in the 10 m and 100 m distances classes on Axis 2 (Fig 4) suggests that this axis may have just explained random species occurrence or survivorship in a zone with few species.

Individual species responses to the distance to the haul road varied considerably as shown both by correlation with Axis 1 and the results of Indicator Species Analysis (ISA). When ISA was first run using each distance class as a group, 94% of the positive ISA associations (where $p < 0.05$) with a given distance class occurred in the range of 1000–4000 m from the road; there were no positive associations in the 10–100 m range (Table 3). Following this pattern, we chose two groups for our final ISA test: < 1000 m and ≥ 1000 m. Species highly correlated with Axis 1 were generally also highly associated in ISA with the ≥ 1000 m group (S4 Table). Twelve species had correlations r > 0.5 on Axis 1 and 11 of these were highly significantly associated with the group of plots ≥ 1000 m ($p < 0.01$). Fifty-one species had significant

**Table 3. Number of indicator species analysis p values < 0.05 in the 7 distance classes from the DMTS haul road in CAKR.**

| Distance Class (m) | Number of ISA p-values < 0.05 |
|---|---|
| 10 | 0 |
| 50 | 0 |
| 100 | 0 |
| 300 | 2 |
| 1000 | 7 |
| 2000 | 10 |
| 4000 | 14 |

The MRPP A-statistic, a measure of the strength of community differences in multivariate species-space, showed two general clusters of plots—those between 10–100m and those ≥ 1000 m (Table 4). The 300 m plots showed weak differences between each of these groups. Plots at the 1000, 2000 or 4000 m distances showed only weak lichen community differences among themselves (A = 0.02) but strong to moderate differences with plots in the 10–100 m distance group (0.05 ≤ A ≤ 0.20). A-statistics were the highest in comparing the 2000 m or 4000 m plots with those close to the road.

associations with this indicator group (p < 0.05), suggesting that over half of the species were sensitive to the effects of fugitive dusts, and at a range of sensitivities. There were no species

**Table 4. MRPP A-statistics from pairwise comparisons plot distance classes for lichen, vascular plant and bryophyte communities.** Color shadings reflect groupings of strong (red), moderate (yellow) and weak (blue) or little (black) community differences. Community differences were evaluated using Sorenson distance measure on $\log_{10}$ transformed matrices of: lichens (91 plots x 93 species), vascular plants (91 plots x 56 species) and bryophytes (91 plots x 5 species groups).

| Distance Class (m) | 10 | 50 | 100 | 300 | 1000 | 2000 | 4000 |
|---|---|---|---|---|---|---|---|
| **LICHEN COMMUNITIES** | | | | | | | |
| 10 | - | 0.05 | 0.09 | 0.14 | 0.16 | 0.2 | 0.18 |
| 50 | | - | <0.01 | 0.04 | 0.08 | 0.1 | 0.11 |
| 100 | | | - | <0.01 | 0.05 | 0.07 | 0.08 |
| 300 | | | | - | 0.01 | 0.03 | 0.03 |
| 1000 | | | | | - | 0.02 | 0.02 |
| 2000 | | | | | | - | <0.01 |
| 4000 | | | | | | | - |
| **VASCULAR PLANT COMMUNITIES** | | | | | | | |
| 10 | - | <0.01 | <0.01 | <0.01 | 0.02 | 0.07 | 0.06 |
| 50 | | - | <0.01 | <0.01 | <0.01 | 0.06 | 0.04 |
| 100 | | | - | <0.01 | 0.02 | 0.05 | 0.03 |
| 300 | | | | - | <0.01 | <0.01 | <0.01 |
| 1000 | | | | | - | <0.01 | <0.01 |
| 2000 | | | | | | - | <0.01 |
| 4000 | | | | | | | - |
| **BRYOPHYTE COMMUNITIES** | | | | | | | |
| 10 | - | <0.01 | 0.08 | 0.14 | 0.15 | 0.26 | 0.25 |
| 50 | | - | 0.07 | <0.01 | 0.08 | 0.2 | 0.19 |
| 100 | | | - | <0.01 | <0.01 | 0.1 | 0.07 |
| 300 | | | | - | <0.01 | 0.05 | <0.01 |
| 1000 | | | | | - | <0.01 | <0.01 |
| 2000 | | | | | | - | <0.01 |
| 4000 | | | | | | | - |

that had a significant association with the 10–300 m group at the p < 0.05 level, suggesting that the dusts did not enhance the habitat for any taxa.

### Spatial analysis of LSR

The log of modeled LSR was strongly correlated with the log of distance to the haul road ($r^2$ = 0.89, Fig 6) in the posterior predictions analysis of 1008 prediction points. Similarly to the distribution of Zn, Pb, and Cd reported in Neitlich et al. [17], the posterior prediction modeling of LSR showed strong increases away from the road and port-based contaminant sources (Fig 7). In the raw data, LSR ranged from 0 adjacent to the road to 56 at 4000 m. In the model LSR ranged from 2.6 to 54.7. Most of the area out to 1000 m from the road had a LSR lower than 75% of that of the mean LSR (40.8) of the 2000–4000 m stratum.

When summarized by distance class, the LSR of each distance class from 3000 m to the haul road had a 100% probability of having a higher LSR than the next closest class to the road (Table 5). The 3000–4000 m class had an 86% probability of having a higher LSR than the 2000–3000 m class. In short, the model summarization showed a 100% probability of impacts to LSR out to 3000 m and very high probability of impacts out to 4000 m. Since we didn't measure LSR beyond 4000 m, we could not make any inferences about points beyond this distance.

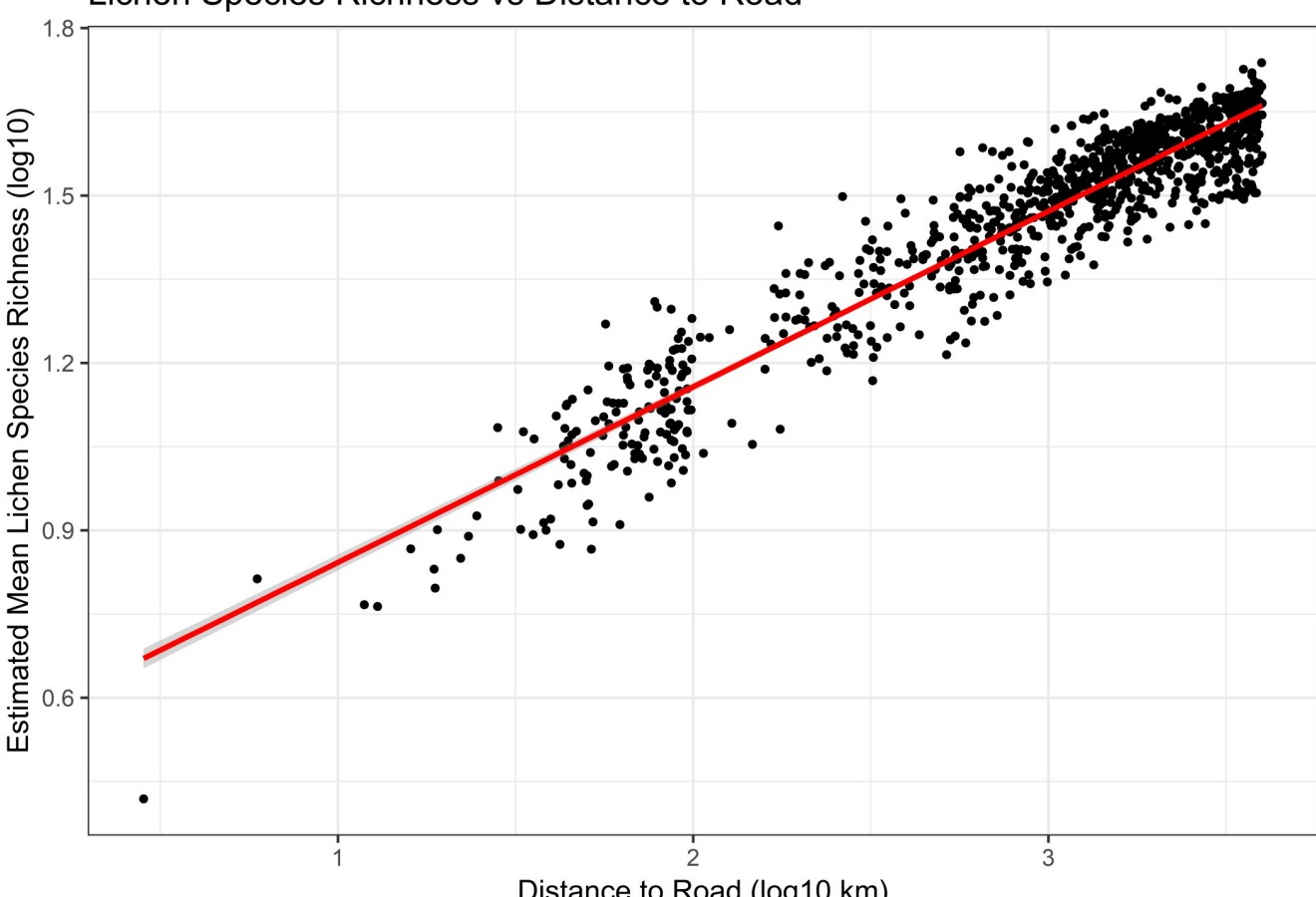

**Fig 6. Estimated mean lichen species richness vs. distance to the DMTS haul road in Cape Krusenstern National Monument, Alaska, on a log-log scale from a Bayesian posterior predictions model.** The 95% confidence interval is shown in grey.

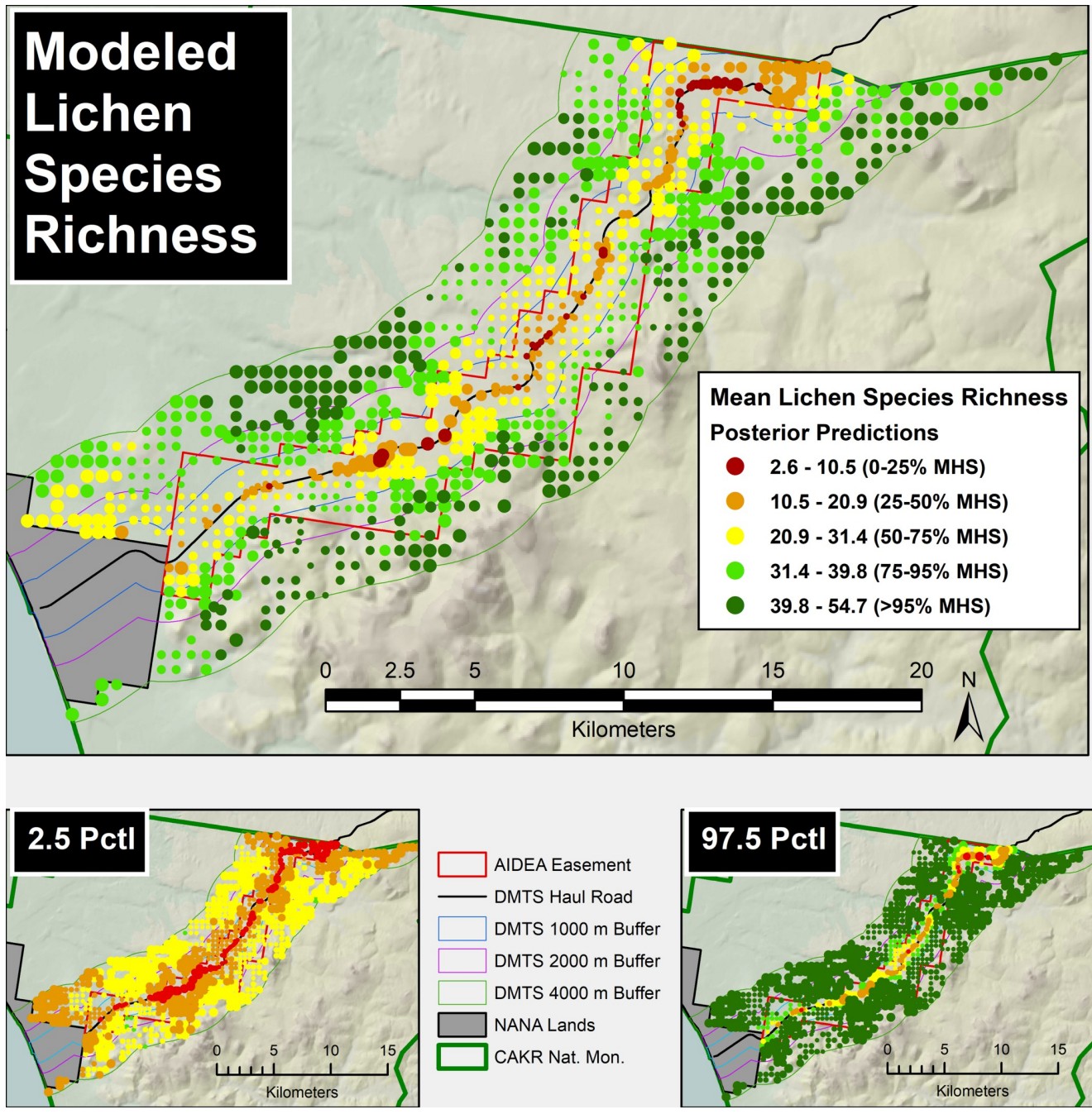

**Fig 7. Modeled lichen species richness in a prediction grid from 0 to 4000 m from the DMTS haul road in Cape Krusenstern National Monument, Alaska.** Prediction grid points are limited to those in the same land cover types as the study plots, as other land cover types have different lichen diversity and cover. Road buffer lines are drawn at 10, 50, 100, 300, 1000, 2000 and 4000 m. Points are sized proportionally in four classes by the quartile distributions of the reciprocal of the coefficient of variation (CV). The upper and lower bounds of the 95% confidence interval (i.e., 97.5 and 2.5 percentile values of the 200 simulations) are provided in small maps below the main map.

The LSR was considerably lower within the NANA industrial road easement than in CAKR proper (Table 6 and S1 Fig). Approximately 22% of the study area fell inside of the easement. A total of almost 19 km$^2$ outside of the easement (or approximately 9% of the total area studied outside of the easement) had an LSR ≤ 75% of that in the highest distance class, whereas a

**Table 5. Lichen species richness (LSR) model summarized by distance class from the DMTS haul road in CAKR.** Summarization methods for mean, standard error (SE), minimum, maximum, 2.5 percentile, 97.5 percentile and probability of decrease (Pr Decrease) follow those used in Neitlich et al. [17] for contaminants. The probability of decrease (Pr Decrease) reports the percentage of decreases in the comparisons of the 200 posterior predictions of each adjacent distance class.

| Distance Class | Distance to Road (m) | N | Mean LSR | Mean SE LSR | Min LSR | Max LSR | 2.5 PCTL Mean LSR | 97.5 PCTL Mean LSR | Pr Decrease from Next Farther Distance Class |
|---|---|---|---|---|---|---|---|---|---|
| 1 | 0–50 | 37 | 9.4 | 1.0 | 2.6 | 13.6 | 7.7 | 11.4 | 100 |
| 2 | 50–100 | 103 | 13.2 | 1.1 | 7.4 | 20.4 | 11.5 | 15.8 | 100 |
| 3 | 100–300 | 51 | 19.1 | 1.2 | 10.9 | 31.5 | 17.0 | 21.6 | 100 |
| 4 | 300–1000 | 196 | 26.6 | 1.9 | 14.7 | 39.5 | 23.7 | 31.3 | 100 |
| 5 | 1000–2000 | 267 | 34.6 | 2.5 | 22.8 | 46.6 | 30.4 | 40.1 | 100 |
| 6 | 2000–3000 | 175 | 39.5 | 3.1 | 27.8 | 49.5 | 34.7 | 46.5 | 86 |
| 7 | 3000–4000 | 179 | 41.9 | 3.5 | 31.1 | 54.7 | 34.4 | 48.6 | - |

From the Neitlich et al. [17] prediction grid used in Fig 7, we summarized the area of prediction points in the two land cover types most likely to have highest lichen cover. The 3000–4000 m distance class was the class with the highest modeled LSR (41.9). Over 10 km$^2$ (or 4% of the study area out to 4000m) had a LSR of $\leq$ 50% of the mean LSR of the 3000–4000 m distance class (Table 6). A total of nearly 55 km$^2$ (21% of the study area) had a LSR of $\leq$ 75% of the 3000–4000 m distance class. A total of 157 km$^2$ (59% of the study area) had a LSR of $\leq$ 95% of the 3000–4000 m distance class, while the remaining 41% of the study area had a LSR of > 95% of that class.

total of 36 km$^2$ inside of the easement (or approximately 62% of the total area inside the easement) had the same LSR. Approximately 91% of the area outside the easement had an LSR of $\geq$75% of that in the highest distance class (with 42% of the area outside the easement at 75–95% of the highest class's LSR and 49% of the area at $\geq$ 95% of the highest class's LSR). By contrast, inside the easement only 38% of the total area had an LSR of $\geq$ 75% of that in the highest distance class. Only 11% of the total area inside the easement had an LSR $\geq$ 95% of the highest distance class vs. 49% outside the easement. There were several areas outside the easement in which the LSR was $\leq$ 75% of that of the highest class; the two areas in which LSR outside the easement appears to be lowest are to the north of the Port Site and north of the road towards the middle of the monument (Fig 7). Consistent with prevailing wind direction and a general skew of the easement toward the south of the haul road, more of the areas outside of the easement showing surpressed LSR $\leq$ 75% of the mean high stratum occurred north of the road.

## Vascular plant communities

A log-transformed matrix of 93 plots x 36 vascular plant species was ordinated with NMS numerous times using the same settings as in the lichen ordination. The chosen ordination

**Table 6. Area estimates in CAKR by percentage of the mean LSR of the 3000–4000 m distance class (41.9) according to the posterior predictions model.** The areas represented here are only those within the study area (i.e., within a 4000 m buffer of the DMTS haul road and on NPS lands).

| | ENTIRE STUDY AREA | | | | | | INSIDE vs OUTSIDE AIDEA EASEMENT | | | |
|---|---|---|---|---|---|---|---|---|---|---|
| Percent of Highest Distance Class's Mean LSR | LSR Range | N | Area (km$^2$) | Cumulative Area (km$^2$) with Lesser or Equal LSR | Percent of Area | Cumulative Percent of Area with Lesser or Equal LSR | Area Inside AIDEA Easement (km$^2$) | Area Outside AIDEA Easement (km$^2$) | % of Total Area Inside AIDEA Easement | % of Total Area Outside AIDEA Easement |
| 0–25 | 2.6–10.5 | 37 | 0.3 | 0.3 | 0.1 | 0.1 | 0.3 | 0 | 0.5 | 0 |
| 25–50 | 10.5–20.9 | 160 | 10.4 | 10.7 | 3.9 | 4 | 9.9 | 0.3 | 17.2 | 0.2 |
| 50–75 | 20.9–31.4 | 237 | 43.8 | 54.6 | 16.5 | 20.5 | 25.4 | 18.5 | 44 | 8.9 |
| 75–95 | 31.4–39.8 | 334 | 102.6 | 157.2 | 38.5 | 59 | 15.6 | 87.1 | 27.1 | 41.8 |
| $\geq$95 | 39.8–54.7 | 240 | 109.1 | 266.2 | 41.0 | 100 | 6.4 | 102.7 | 11.1 | 49.2 |
| Total | - | 1008 | 266.2 | - | 100 | - | 57.6 | 208.6 | 100 | 100 |

was rotated to align Axes 1 and 2 maximally with the most highly correlated variables and to obtain 100% orthogonality. The final configuration explained 85% of the variance with 42% on Axis 1, 18% on Axis 2 and 26% on Axis 3 (Fig 8). Axis 1 was moderately correlated with the log of the distance to the haul road (r = 0.44), vascular plant species richness (VPSR, r = 0.53) and LSR (r = 0.38). This axis was most negatively correlated with the modeled 2001 Cd, Pb and Zn values (r = –0.46, –0.44 and –0.44 respectively) from Hasselbach et al. [18]. Axis 1 correlated weakly with the 2006 values for these same elements (r = –0.27, –0.24, and –0.22 respectively). Correlations of crustal elements with this axis were similarly low.

Axis 2 of the vascular ordination showed weak positive correlation with VPSR (r = 0.35) and with the cover of *Hylocomium splendens* (r = 0.29), which was most abundant in mesic environments not dominated by *Eriophorum vaginatum*. There were no variables with a strong negative correlation with Axis 2. Axis 3 had no strongly correlated variables and was of little interpretive value. The vascular plants most positively correlated with Axis 1 were *Carex bigelowii* (r = 0.83), *Pyrola* sp. (r = 0.53) *Salix reticulata* (r = 0.46), and *Saxifraga punctata* (r = 0.46). These plants are generally associated with more mesic conditions on a local scale, though elevation per se or landscape position had extremely low correlations (r < 0.1) with all axes. The species most negatively correlated with Axis 1 were *Carex* sp. (r = –0.66) and *Ledum decumbens* (r = –0.62), which tend occur in wetter conditions.

In the NMS joint plot (Fig 8) there was a higher proportion of plots in the 10 m to 100 m distances from the haul road on the negative side of Axis 1, but far weaker ordinal clustering by distance than in the lichen ordination. Paired comparisons of distance classes in MRPP also showed weaker differences in vascular plant community structure at most distances than those of the lichen communities (Table 4). Six distance pairs on the transect plots showed strong community differences (A≥ 0.10) for lichen communities versus none for vascular plants and

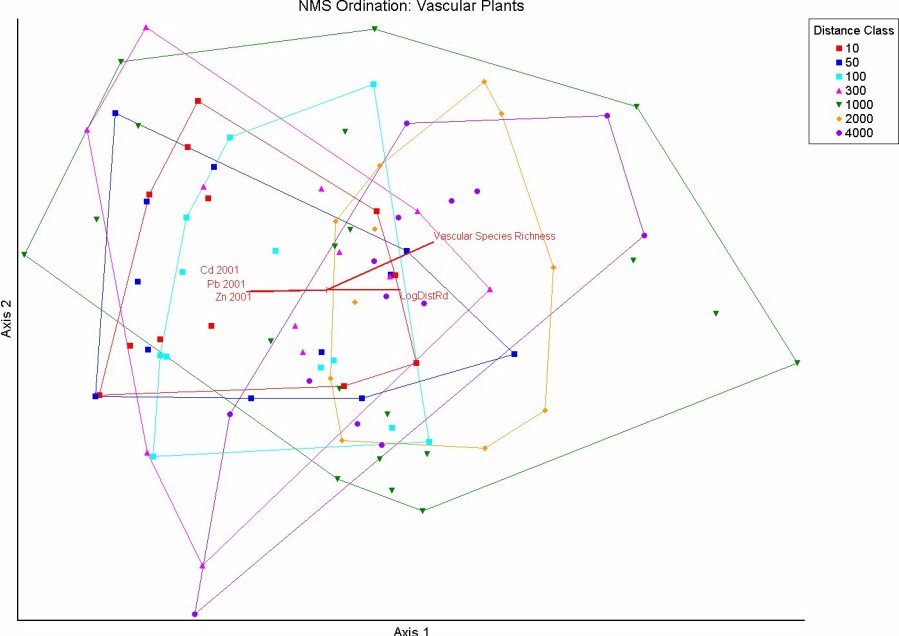

**Fig 8. Nonmetric multidimensional scaling ordination of vascular plants in a matrix of 93 plots x 36 species along the DMTS haul road in Cape Krusenstern National Monument, Alaska.** The ordination is overlain by vectors proportional to the strength of their correlation with ordination axes. Convex hulls surround plots in each distance class. Variables with Spearman correlations r > 0.4 are presented: LogDist_Rd (log$_{10}$ distance to the road), Vascular Species Richness (VPSR), and modeled Cd, Pb and Zn from 2001 [18].

seven for bryophytes. The mean A-statistic scores for vascular community comparisons that had significant results were 53% as large as the lichen community comparisons. The vascular plant comparisons had 13 non-signficant results versus only 3 for lichen comparisons.

## Bryophyte communities

A final $log_{10}$-transformed bryophyte matrix of 94 plots by 5 species groups relativized by species group maximum yielded a two-dimensional solution that explained 87 percent of the variance in the data set. Fifty percent of the variance was loaded on Axis 1, with 37 percent on Axis 2 after rotation with the most highly correlated variable, the log of distance to the road. Despite the loss of a large amount of information due to the grouping of taxa into broad composite entities, NMS ordination for bryophytes yielded similar though weaker results than that for lichens (Fig 9).

Axis 1 was most highly correlated with $log_{10}$ distance to the haul road (r = 0.652), $log_{10}$ LSR (r = 0.554), and LSR (r = 0.515, Table 2). The most highly negatively correlated variables with Axis 1 were the elemental tissue contents of *Hylocomium splendens* of 14 elements ranging from -0.671 for Cr to -0.561 for Total S. Al, Ca, Cd, Fe, Pb and Zn all fell in this range. The correlations (r) of modeled values of Cd, Pb and Zn from 2001 with Axis 1 ranged from –0.512 to –0.561. Axis 2 was weakly negatively correlated with the cover of *H. splendens* (r = –0.442) and vascular plant cover (r = –0.305) and had no positive correlations with r > 0.17.

*Sphagnum* spp. were highly correlated with Axis 1 (r = 0.862) and *Hylocomium splendens* was weakly negatively correlated with this axis (r = –0.404). *H. splendens* was strongly negatively correlated with Axis 2 (r = –0.755) and pleurocarpous mosses were moderately negatively correlated with this axis (r = –0.565). In ISA, *Sphagnum* was strongly associated with the 1000–4000 m from the haul road distance class in every test grouping (Table 7) and was most highly associated with the 2000 m distance class in particular when analyzed with all 7 transect distance classes. Liverworts, which had a correlation of r = 0.368 with Axis 1 also had a significant association with the 1000–4000 m and 2000 m distance classes in ISA tests. Taken together, these results suggest that *Sphagnum* and liverwort cover was reduced at distances less than 1000 m and the natural community structure of bryophytes may have persisted intact at ≥ 2000 m from the road. Given the high indicator values of both acrocarpous and pleurocarpous mosses (with the exception of *H. splendens*) at distances closer to the road, it is likely that *Sphagnum* in near-road locations is being replaced by these mosses. Analysis of species richness patterns would require a full bryophyte survey.

In MRPP (Table 4), the full bryophyte community matrix showed the highest community differences between distance classes from the haul road of any ecosystem component (A = 0.26 in comparing 10 vs. 2000 m and A = 0.25 in comparing 10 vs. 4000 m). Bryophytes had a higher number of strong differences (7) in transect plot pair comparisons than lichens or vascular plants, but also had a relatively high number of non-significant differences (9) on transect plot pairwise comparisons compared to the lichen matrix (3). Both the 10 and the 50 m distance classes showed maximum community differences with the 2000 and 4000 m distance classes. Differences in community structure for bryophytes appeared to be more intense than that for lichens, but the results are not directly comparable because the bryophyte analysis is based on broad species groups rather than individual taxa.

A joint ordination of vascular plants, lichens and bryophytes showed weaker relationships than the lichen analysis because the strong lichen signal was diluted by a large input of taxa with no strong relationship to road distance.

## Nonvascular plant vigor

In the linear models used for these metrics, the mean percent of the *Hylocomium splendens* midrib turning black at the 10 m (72%) and 50 m (24%) distances from the haul road was significantly different from those at farther distances, where the means ranged from 1 to 10% (Table 1, F = 48.3, p < 0.001). $Log_{10}$ percent of the *H. splendens* midrib turning black had a negative correlation with $log_{10}$ distance to the road ($r^2$ = 0.55) and with elemental concentrations (S3 Table). Frond widths ranged from 1.3 cm to 1.6 cm over the study area, and only the

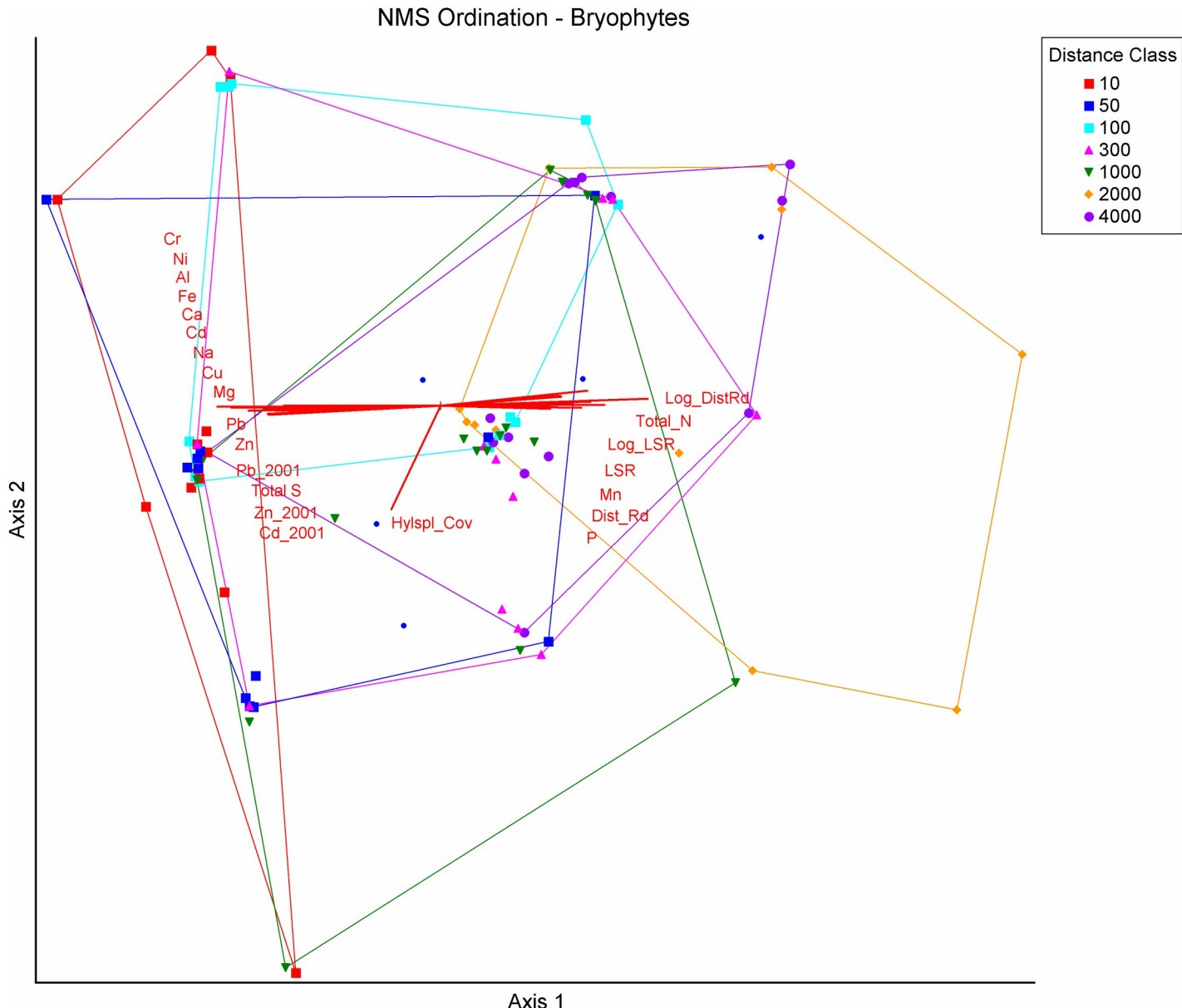

**Fig 9. Nonmetric multidimensional scaling ordination of bryophytes on a matrix of 94 plots by 5 species groups along the DMTS haul road in Cape Krusenstern National Monument, Alaska.** The ordination is overlain by vectors proportional to the strength of their correlation with ordination axes. Spearman r values for each variable are provided in Table 2. Convex hulls surround plots in each distance class. All two letter abbreviations are elements, and those followed by "_2001" are modeled values from 2001 sampling [18]. Mean elemental values were mg/kg in *Hylocomium splendens* tissue, dry weight, except % Total S and % Total N. Other abbreviations are: Total_S (Total sulfur), Hylspl_Cov (*H. splendens* cover), LogDist_Rd ($log_{10}$ distance to the road), Total_N (Total nitrogen), Log_LSR ($log_{10}$ lichen species richness), LSR (lichen species richness), P (phosphorus), Dist_Rd (Distance to the road), P (phosphorus).

**Table 7. Indicator species analysis of bryophyte taxon groups by distance class groups from the DMTS haul road, CAKR.** Only results with p values < 0.05 are shown below. *Hylocomium splendens* does not appear as all indicator values for this group had p values > 0.05. Tests were run for each distance class individually and for 10–300 m vs. 1000–4000 m.

| Bryophyte Taxon Group | Number of ISA Groups in Test | ISA Groups in Test | Indicator Group of Maximum Value | p |
|---|---|---|---|---|
| Acrocarpous moss | 2 | 10–300, 1000–4000 | 10–300 | 0.0466 |
| Liverwort | 7 | 10, 50, 100, 300, 1000, 2000, 4000 | 2000 | 0.0306 |
| Liverwort | 2 | 10–300, 1000–4000 | 1000–4000 | 0.0238 |
| Pleurocarpous moss | 7 | 10, 50, 100, 300, 1000, 2000, 4000 | 50 | 0.0382 |
| Sphagnum moss | 7 | 10, 50, 100, 300, 1000, 2000, 4000 | 2000 | 0.0064 |
| Sphagnum moss | 2 | 10–300, 1000–4000 | 1000–4000 | 0.0002 |
| Sphagnum moss | 3 | 10–50, 100–300, 1000–4000 | 1000–4000 | 0.0002 |

mean frond width at 10 m (1.3 cm) was significantly different from the other distance classes, which ranged from 1.4–1.6 cm (F = 2.8, p < 0.01).

There was a moderate positive correlation ($r^2 = 0.39$) between the $\log_{10}$ of the distance from the haul road and the mean heights of the five lichens for which height was measured: *Flavocetraria cucullata*, *Cetraria laevigata*, *Cladina arbuscula/mitis*, *Cladina stygia*, and *Thamnolia subuliformis/vermicularis*. *Cetraria cucullata* and *Cetraria laevigata* had a slightly higher correlation with $\log_{10}$ of the distance from the road ($r^2 = 0.41$ for each).

The total % nitrogen (N) in moss tissue reflected these changes in plant health. Unlike every other element that had a moderate to strong correlation with distance to the road, Total N was positively correlated ($r^2 = 0.59$) with $\log_{10}$ of distance from the haul road. Mean % total N had a positive correlation with distance from the road and ranged from 0.62% at 10m from the road to 0.99% at the reference sites. Percent total N became statistically indistinguishable from background levels at distances $\geq$ 1000 m from the road (F = 23.9, p < 0.0001). Presumably, total N levels related to the health of the moss tissue in that dying or compromised tissue closer to the road could not retain the optimal levels of N.

## Discussion

### Context within other lichen-based pollution studies

While this study reiterates the conclusions of numerous studies finding that lichen communities are sensitive to pollution, it differs from many of these studies in certain respects. Frequently, ordination analysis relating vegetation community structure to pollution have included large areas (e.g.,[51, 52]) and have required careful selection of data inputs such that the key portions of the gradients of interest were both well-covered and not over-represented [52–54]. Many pollution studies have struggled with the confounding effects of large geographic extent, vegetation type, multiple lichen community types, ecoregion, climatic variability, elevation, topography, and multiple or mobile pollution sources. This study, by contrast, represents a tidy, spatially-confined case study demonstrating the effects of a linear point source set in an extremely clean area, with few confounding factors aside from the variety of co-dispersed metals present. By arraying sample units on a log-based contaminant decay curve, this study had the luxury of pre-balancing ordination inputs along pollution gradients. And, by limiting sampling to two related land cover types within a narrow elevation band and a single resultant lichen community type, we eliminated much of the noise common to lichen air pollution studies. Because of the almost experimental nature of placing a road-based pollution source in a pristine environment with limited environmental variability other than pollution levels, we present in this study a variety of approaches to illuminate different aspects of vegetation community structure.

In most lichen air pollution studies, LSR can be important (as pollution eliminates certain species) but does not usually by itself explain the bulk of the relationship between community structure and pollution. Frequently, these studies rely for explanatory power on the subsets of lichens that are differentially tolerant of pollutants including nitrogen, acidic deposition, and heavy metals [11, 25, 51, 53], and in the case of nitrogen, even a subgroup enhanced by this pollutant [54]. As such, LSR by itself is not usually as good an indicator of pollution gradients as community composition and subgroup membership. By contrast, in this study, LSR emerged as the single most important variable related to pollution and explained most of the pollution gradient by itself. Because of the strength of LSR's correlations (r = 0.96 on ordination Axis 1 and $r^2$ = 0.99 vs $\log_{10}$ distance to the road in posterior predictions), we were able to treat LSR as a direct measure of the pollution gradient. ISA results provided some insight into why this was the case: the overwhelming majority of lichen species in this study were associated with distances of at least 2000 m from the road, and none were significantly associated with distances less than 300 m. By eliminating a large number of taxa close to the road, fugitive dusts created a strong LSR gradient based on proximity to the road and port site. Lichen cover, while diminished greatly at distances of less than 300 m from the road, had higher variability at greater distances and thus proved less reliable as a pollution indicator on a landscape level.

There are a few other key differences between this study and most other lichen pollution studies. First, while many studies address the effects of pollution from smelters on vegetation, and many use lichens and mosses as biomonitors of contaminant levels, very few have studied the effects of mining-related fugitive dusts on vegetation. This difference between the elemental or oxidized state of smelted metals versus the reduced state of metal sulfides in this study make direct comparison of pollution effect levels challenging. The current study will prove most useful in evaluating pollution effects onsite as levels change due to mitigation or closure, and in analogous arctic environments with mining haul roads. Second, due to the small spatial scale of this study and the potential for improvement of contaminant levels with additional operational controls and the eventual closure of the facility many decades from now, this study serves as a better baseline for monitoring recovery than most studies situated in areas of likely continued impact.

## Estimating background LSR

While most studies of lichens and air pollution have used ordination and a variety of modeling approaches for large and heterogeneous study areas [51, 53, 54], spatial models can be useful in account for the high spatial autocorrelation inherent in fine-scale studies of road-based pollution. Like other studies of road dust effects on vegetation (e.g., [26, 27]) we used conventional tools of vegetation ecology, but supplemented them with spatial modeling to improve our geographic inference. We evaluated multiple lines of evidence in determining baseline condition for lichens and other vegetation through ordination, MRPP, ISA, linear models and modeling of the most important variables via Bayesian posterior prediction. Since LSR was the biological variable by far most correlated with the ordination lichen gradient associated with pollution, we based our posterior predictions spatial model on that variable. Analysis by distance-based strata suggested that there was a 100% chance that the LSR between 2000–3000 m was higher than that in the next closest stratum (1000–2000 m), and that there was an 86% chance that the LSR between 3000–4000 m was higher than that in the 2000–3000 m stratum (Table 5). This placed high certainty on an effects threshold of out to 3000m and reasonably high probability that LSR continued to increase out to 4000 m, the maximum distance sampled and modeled. Due to the incompatibility of our reference sites with transect sites, we could not make any inferences about background LSR levels beyond 4000 m.

Mapping of the spatial model (Fig 7) provided geographic insights that highlighted the hot-spots of LSR depression near the Port Site and the northern monument boundary. Because of the challenge of creating thresholds in continuous variables, we used reasonable LSR classes based on percentages of the mean LSR of the highest stratum (3000–4000 m). The darkest green dots in Fig 7 depict the cleanest areas in the study area (with ≥ 95% of the mean LSR of the highest stratum), but again we were unable to conclude that this stratum represented an LSR background level. In these relatively clean locations, other factors such as microsite topography, wetness, and vascular plant variability were likely to have exerted more influence on LSR than pollution. The mapping of modeled LSR is analogous to other efforts to map lichen community outputs based on the most significant community attribute aligned with pollution vectors after the ordination has been rotated accordingly [52, 53].

Other research confirms the basic patterns of our findings but with less taxonomic detail. In 2007, Exponent, Inc. [20] conducted vegetation studies along the haul road and found that lichen cover was significantly reduced (2 to 4.5 fold) out to 2000 m compared to reference site covers. Their studies did not use species distinctions for lichens and bryophytes (only broad classes), but they concluded that lichen cover, lichen frequency, moss cover and dwarf shrub cover were all reduced at sites close to the road (10 m and 100 m distance) relative to 1000–2000 m distance or reference sites. Their ordinations included all vascular taxa and single entries for lichens and mosses in all landscape settings and distances from the road. This allowed them to separate out vegetation types that varied widely with physiography but permitted no inference on LSR gradients or differences in sensitivity between vegetation components. Because of the variety of vascular plant habitats sampled, differences between community types formed stronger gradients than those related to pollution. Both Nash [25] and Folkeson and Bringmark [11] noted that Zn (with the addition of Cd or Cu) had the effect of impoverishing LSR and cover around smelting operations. Tømmervik et al. [10] noted widescale damage to lichen communities in Norway due to Russian smelting in the border region, but did not explore the fine-scale details of the taxonomic nature of the damage.

### The role of crustal dust

One of the key questions emerging from our results is the extent to which lichen species impoverishment along the haul road may be attributed to the crustal elements (e.g., Al, Fe, Ca, Mg) or physical properties of road dust vs. the Zn, Pb and Cd enrichment documented in Neitlich et al. [17]. It is likely that dust at levels deposited near the road may impact the physiology of both nonvascular and vascular taxa in terms of photosynthesis, respiration, stomatal conductance, water uptake and nutrient status. Several studies have reported effects on lichens in arctic tundra environments from crustal road dust along the Dalton Highway (or Prudhoe Bay haul road, approximately 575 km E of the haul road). Walker and Everett [55] found that lichens were the most affected component of plant communities in roadside environments and were killed in high dust areas. Lichens did not regain background characteristics until at least 70 m from the road. *Sphagnum* moss was also eliminated in these roadside environments. Auerbach et al. [26] found that plots immediately adjacent to the road had significantly higher pH, higher biomass of graminoids and *Rubus chamaemorus*, and significantly lower biomass of lichens, mosses, forbs and dwarf shrubs than distant sites out to 1000 m. A brief exploration of the lichen data in their study suggested that the worst effects of crustal road dust on LSR occurred at less than 50 m from the road (S1 File). Myers-Smith et al. [27] studied the effects of road dust on the Dalton Highway on vegetation after 25 years of road operation. While data on lichen species was not gathered, they found that the major response of vegetation composition to road dust was an increase in graminoids and a decrease in *Sphagnum* spp. and lichens.

Lichen cover and biomass appeared to be reduced dramatically at distances less than 100 m, and effects were detected beyond 100 m (but less than 400 m) from the road. Effects also increased at distance with time: while early impacts from road dust (<15 years) were limited to 50 m, by 25 years effects extended beyond 100 m. An extensive literature review by Farmer [56] noted that the most damaging effects of road dust on any natural community were in arctic systems due to the high presence of lichens and bryophytes.

Additional support for the minor reduction of LSR via crustal road dust vs heavy metals comes from 3 plots on a very dusty gravel road in Kotzebue, Alaska. The mean LSR from these plots, located at 10, 50 and 100 m from the road was 34, which was equivalent to modeled LSR in the 1000–2000 m stratum on the haul road (Table 5) and the raw data (Table 1). In MRPP, the Kotzebue Road plots showed a high level of difference compared to roadside plots at the same distances on the haul road ($0.09 \leq A \leq 0.18$), presumably because of their much higher LSR and cover.

Taken in sum, the results of this study suggest that the effects of fugitive dusts on lichens out to 4000 m from the road are unlikely to be the result merely of crustal dust, and are most likely to be involve effects from Zn, Pb, Cd and S in these dusts.

## Heavy metal effects

Most of the studies examining the relationship between heavy metals and lichens have occurred near smelters [e.g., 10, 11, 25]. Comparisons between dust-borne contaminants in this study and the aerosols often found at smelters should be viewed with caution as gas-phase compounds are more reactive and typically cause more tissue damage [57]. Moreover, because smelter emissions tend to release metals in their elemental states—rather than the reduced metal-sulfide states accompanying Zn and Pb ore concentrates at Red Dog—comparisons between those numerous studies and ours may not be fruitful. Field conditions close to smelters typically feature the combined effects of a suite of metals, sometimes in concert with sulfates [58]. Tyler [58] suggests that the relative toxicity of common heavy metals to lichens decreases in the order Hg, Ag > Cu, Cd (?) > Zn, Ni >Pb. However, the degree of sensitivity varies considerably within populations, and individuals may increase in their tolerance to sublethal metals concentrations metals over time.

The toxic effects of Zn on lichens are well-known [11, 25]. In this study, lichens close to the road feature a broad mix of stressors including crustal elements, Zn, Pb, sulfides, and the physical effects of smothering by dust. Lichens at greater distances were exposed to Zn, Pb, Cd and a suite of crustal metals about an order of magnitude lower than their levels close to the road. Our working hypothesis is that while lichens were most affected by the combination of heavy metals and crustal dust at distances less than 100 m, they were most affected by Zn (and to a lesser degree Pb and sulfides) at distances out to 3000 m or beyond. Nash [25] concluded that Zn and Cd were equivalently toxic to lichens. Levels of Cd along the haul road corridor are typically two orders of magnitude below that of the other main stressors, thus Cd is much less likely to play a major role in the reported lichen decline. While Zn was slightly less highly correlated with the lichen ordination than crustal elements, the diffusion of many elements onto the landscape is highly cross-correlated, making it challenging to partition effects without experimental evidence. A spatial analysis of contaminant thresholds associated with biologically significant markers (e.g., first reductions in LSR and cover, mortality) could potentially help untangle this knot.

Still, on a local scale, if we treated Zn as a proxy for a cocktail of elements (and assumed that the proportion of each element was relatively static), we could use this element as an indicator. By overlaying the spatial model of contaminants [17] onto the LSR model, we find that

the concentrations of Zn associated with mapping or stratum-based estimates of likely LSR backgrounds ranged from about 58–70 mg/kg in moss tissue (S2 File). The correlation of LSR vs. Zn between the two models was strong ($r^2 = 0.77$, $p < 0.001$, S2 File), though certainly most other metals would have similar values if modeled spatially and regressed against LSR as these metals are highly cross-correlated in linear models (S2 Fig and S3 Table). Keeping Zn levels in *Hylocomium splendens* tissue below this level would be likely to reduce impacts to lichen communities, regardless of varying elemental effect levels and/or synergistic effects between elements.

## Effects relative to the easement

Under the terms of ANILCA, CAKR was established "to protect habitat for and populations of, birds, and other wildlife, and fish resources; and to protect the viability of subsistence resources." In 1984 Congress granted NANA a 99-year road easement through CAKR via Public Law 99–96 USC 43.1629. While the U.S. Government retains ownership of the easement, the land may be used by the Alaska Industrial Development and Export Authority (AIDEA) as desired but within the terms of an easement agreement that protects fish, wildlife and habitat during mining operations. Most of the impacts to lichens fall within the NANA easement (Table 6), and it is unknown whether these impacts are within the scope of the agreement. This study also suggests, however, that vegetation on a significant amount of land in CAKR but outside of the industrial easement has been affected by fugitive dust emissions. Our model indicates that as of 2006 on lands outside the easement, approximately 19 km$^2$ may harbor only 50–75% of its potential number of lichen species, and another 87 km$^2$ may harbor 75–95%. These numbers may represent an underestimate as the mean modeled LSR of the highest stratum may be lower than that of an appropriate reference site, had one been found. It is apparent from Fig 7 that the bulk of these diminished LSR values are on the north side of the road (where contaminants are pushed by prevailing winter easterlies [17], and many occur close to the port site—where soil concentrations of Zn in 2003 were up to 15,000 mg/kg, vegetation was dead or highly stressed, and moss tissue reached levels over 8,000 mg/kg [20]. In plotting LSR vs distance to road, it becomes evident that there is a large "mixing zone" in easement status in an area of reduced LSR (i.e., below 1.6 logLSR in S1 Fig). A great deal of easement status mixing (i.e., the joint presence of grid cells attributed as either inside or outside of the easement) occurs at approximately 25 species per plot, which is approximately 60% of the LSR of the mean of the highest stratum.

## Conclusions

Our results suggest that 17 years of emissions of fugitive dusts enriched with Zn, Pb, and Cd created a zone of lichen impoverishment bracketing the 32 km of the DMTS haul road in CAKR. The majority of lichen species were eliminated at distances close to the road ($\leq 100$ m), and our spatial model suggested a 100% probability that background levels of LSR occurred at distances of up to 3000 m from the haul road, with an 85% probability that background LSR levels occurred between 3000–4000 m from the road, the maximum extent of sampling-based inference. Lichen cover was markedly reduced at distances less than 300 m from the road. NMS ordinations of lichen communities showed a strong association of community structure (driven primarily by LSR) with heavy metal concentrations and distance to the road. Dust emissions also affected bryophyte communities, primarily in replacing *Sphagna* with acrocarpous and pleurocarpous mosses. Vascular plant communities showed only modest response to fugitive dusts.

Posterior prediction modeling of LSR on a landscape level allowed for visualization of areas of greatest lichen impoverishment. The north side of the road and areas adjacent to the Port Site had noticeably depressed LSR, and over 100 km$^2$ showed some level of LSR improvershment with 19km$^2$ showing $\leq$ 75% of the LSR of the highest stratum. Metrics of plant vigor such as percent of *Hylocomium splendens* moss midrib blackening and lichen height showed a strong associations with distance to the road and contaminant levels.

While the current research provides good insight into the biological responses associated with fugitive dusts, it does not address potential future recovery, causal mechanisms, or contaminant-based thresholds for biological variables. Based on field work in 2017, DiMeglio [59] reported no improvement of LSR even though contaminant emissions had decreased between 2001 and 2006 [17]. Due to the slow growth rates of these nonvascular plants (6.2 mm/yr for *Cladonia stygia* per Holt and Bench, [60], a common species in the study area), it is likely that long-term recovery would take many decades after contaminants decrease additionally following mine closure or more intensive fugitive dust control.

## Supporting information

**S1 Fig. Estimated mean lichen species richness vs. distance to the DMTS haul road in CAKR on a log-log scale from a Bayesian posterior predictions model, classed by easement status.** Prediction points inside of the AIDEA industrial easement are shown in blue, while those outside the easement appear in pink. The 95% confidence interval is shown in grey. A log LSR of 1.5 is approximately 31.6, or 75% of the mean LSR of the highest stratum. (PDF)

**S2 Fig. Elemental concentrations in *Hylocomium splendens* moss (mg/kg) along the DMTS haul road in CAKR graphed with best-fit non-linear curves for each element.** (PDF)

**S1 Table. Percent frequency of macrolichen species found on vegetation plots by distance class from the DMTS haul road in CAKR.** (PDF)

**S2 Table. Mean percent lichen cover in plots within different distance classes from the DMTS haul road in CAKR.** (PDF)

**S3 Table. Spearman correlation matrix (r) of elemental concentrations in Hylocomium splendens and a subset of biological variables from 91 plots along the DMTS, Cape Krusenstern National Monument, Alaska.** Excluded were reference plots, Kotzebue plots and 3 plots for which there were several missing values. Hyl spl: *Hylocomium splendens* moss. Fla cuc: *Flavocetraria cucullata* lichen. Cet lae: *Cetraria laevigata* lichen. (PDF)

**S4 Table. Lichen species Spearman correlations (r) with NMS ordination Axis 1 and Indicator Species Analysis p values comparing two groups: < 1000 m and 1000–4000 m from the DMTS haul road in CAKR.** Species are sorted by Axis 1 r scores. (PDF)

**S1 File. Lichen species richness vs. distance to the Dalton Highway in data from Auerbach et al. [26] on a log-log scale.** (PDF)

**S2 File. Modeled LSR in current study vs. modeled Zn from Neitlich et al. [17] on a log-log scale.**
(PDF)

## Acknowledgments

We are greatly appreciative of assistance from: Dr. William Brumbaugh of U.S. Geological Survey–Columbia Environmental Research Center (USGS-CERC) for oversight of elemental analysis of moss tissue, Dr. James Fairchild of USGS-CERC for chlorophyll analysis, Alex Rupprecht for field work, Sean Patrick for lab work and coordination, Jason and Melissa Hickman for moss sample preparation, and Celia Miller for review of the manuscript and assistance with several figures. Teck provided for housing and logistical support during the project. Dr. Jay VerHoef provided spatial modeling.

## Author Contributions

**Conceptualization:** Peter N. Neitlich.

**Data curation:** Peter N. Neitlich, Shanti Berryman, Anaka Mines.

**Formal analysis:** Peter N. Neitlich.

**Funding acquisition:** Peter N. Neitlich.

**Investigation:** Peter N. Neitlich, Shanti Berryman, Linda H. Geiser, Anaka Mines.

**Methodology:** Peter N. Neitlich.

**Project administration:** Peter N. Neitlich, Shanti Berryman.

**Resources:** Peter N. Neitlich.

**Supervision:** Peter N. Neitlich, Shanti Berryman, Anaka Mines.

**Validation:** Peter N. Neitlich.

**Visualization:** Peter N. Neitlich.

**Writing – original draft:** Peter N. Neitlich.

**Writing – review & editing:** Peter N. Neitlich, Shanti Berryman, Linda H. Geiser, Anaka Mines, Alyssa E. Shiel.

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
