## [Decision Letter · Decision Letter 0]

19 Apr 2022

PONE-D-22-08912Impacts to lichens and other tundra vegetation from heavy metal-enriched fugitive dust on National Park Service lands along the Red Dog Mine haul road, Alaska.PLOS ONE

Dear Dr. Neitlich,

Thank you for submitting your manuscript to PLOS ONE. After careful consideration, we feel that it has merit but does not fully meet PLOS ONE’s publication criteria as it currently stands. Therefore, we invite you to submit a revised version of the manuscript that addresses the points raised during the review process.

Overall, both reviewers and I felt this was a well done paper that makes a significant contribution to the literature. There were no major concerns. There are, however, a number of relatively small edits that can be made to improve the clarity and readability of the manuscript. Reviewer #1 in particular has a long list of suggestions. Please do your best to accommodate these suggestions (and those of reviewer #2) where possible. As for the title, do as you see fit!

We look forward to receiving your revised manuscript.

Kind regards,

Tim A. Mousseau

Academic Editor

PLOS ONE

Journal Requirements:

"This research was funded by the National Park Service’s Arctic Network, Inventory and Monitoring Program and the Western Arctic National Parklands."

"This study was funded by the National Park Service Arctic Network and the Western Arctic National Parklands. The funders had no role in study design, data collection and analysis, decision to publish, or preparation of the manuscript."

4. We note that Figures 1, 2, and 7 in your submission contain [map/satellite] images which may be copyrighted. All PLOS content is published under the Creative Commons Attribution License (CC BY 4.0), which means that the manuscript, images, and Supporting Information files will be freely available online, and any third party is permitted to access, download, copy, distribute, and use these materials in any way, even commercially, with proper attribution. For these reasons, we cannot publish previously copyrighted maps or satellite images created using proprietary data, such as Google software (Google Maps, Street View, and Earth). For more information, see our copyright guidelines: http://journals.plos.org/plosone/s/licenses-and-copyright.

a. You may seek permission from the original copyright holder of Figures 1, 2, and 7 to publish the content specifically under the CC BY 4.0 license.  

Reviewers' comments:

Reviewer's Responses to Questions

**Comments to the Author**

1. Is the manuscript technically sound, and do the data support the conclusions?

Reviewer #1: Yes

Reviewer #2: Yes

2. Has the statistical analysis been performed appropriately and rigorously? 

Reviewer #1: Yes

Reviewer #2: Yes

3. Have the authors made all data underlying the findings in their manuscript fully available?

Reviewer #1: Yes

Reviewer #2: Yes

4. Is the manuscript presented in an intelligible fashion and written in standard English?

Reviewer #1: Yes

Reviewer #2: Yes

5. Review Comments to the Author

Reviewer #1: This manuscript expands upon existing work to assess the effect of fugitive dust on lichen community richness. It expands upon previously published work by focusing primarily on lichen communities and corroborating a spatial model with past results.

The study area, target organisms, and management challenge are highly relevant. The study design provides a good attempt to answer research questions. Data is handled thoroughly and fairly with appropriate statistical methods and species community treatments. The authors place appropriate bounds on their analyses and how it may be applied. They do an excellent job placing their work into context of other similar studies and highlight how this study area, although small, is an excellent case example that allows an interpretation approach near “experimental”.

To the best of my knowledge and experience, I recommend this manuscript for acceptance to PLosOne with minor revisions.

In the attached file, I’ve included general comments requiring standardization throughout parts of the manuscript and specific comments relevant only to one or few lines. I am encouraged by the inclusion of background on this area (i.e., the land ownership and private easement issues) that sometime are omitted in scientific studies even if those issues are what instigated the funding of the research.

Reviewer #2: This title is very wordy, I suggest simplifing the title.

Impacts of heavy metals along the Red Dog road in Alaska.

The current title focuses too much on lichens and not on the other taxa and many readers interested in vascular plants might not bother with an article with the current title.

The differences in response to fugitive dust contaminants among the different vegetation components

170 (i.e., lichens, bryophytes, vascular plants) is unique for this study and should be of interest to a very wide audience.

In the Taxonomy section of the Methods.

It would be clearer to state;

Specimens of known genus and unknown species—and for which all species likely to occur at

335 the site were already present—were lumped with the most common species in the data set.

336 Examples of this included "species with in the" Sphaerophorus and Bryoria :genera:

559 species in plots close to the road was visible to observers and in photos (Fig 5). While most

560 lichens species were eliminated at distances close to the road (10–50 m), those present included:

561 (> 50% frequency) Thamnolia subuliformis, Flavocetraria cucullata, Cladonia amaurocraea,

562 and (33–50% frequency) Cladonia stygia, Cladonia arbuscula, Cetraria laevigata, Cladonia

563 maxima, Cladonia rangiferina, Peltigera aphthosa. These are presumably the most pollutant5

Given the data set it seems that there should be a qualifier of what type of pollutant these lichens species are tolerant of? not all pollutants? just these heavy metals?

Figures are well done. Figure captions are well donce and complete.

6. PLOS authors have the option to publish the peer review history of their article (what does this mean?). If published, this will include your full peer review and any attached files.

Reviewer #1: No

Reviewer #2: **Yes: **Roger Rosentreter

---

## [Author Response · Author response to Decision Letter 0]

16 May 2022

Thanks you kindly for the reviews. I have made extensive changes based on the reviews, and these changes are documented in the file "Response to Reviewers" which contains a comments matrix. We have made every effort to incorporate all suggestions and have found them extremely constructive.

---

## [Editor Report · Decision Letter 1]

31 May 2022

Impacts on tundra vegetation from heavy metal-enriched fugitive dust on National Park Service lands along the Red Dog Mine haul road, Alaska.

PONE-D-22-08912R1

Dear Dr. Neitlich,

We’re pleased to inform you that your manuscript has been judged scientifically suitable for publication and will be formally accepted for publication once it meets all outstanding technical requirements. Congratulations! This is a highly rigorous study that significantly moves the field forward. Nicely done!

Kind regards,

Tim A. Mousseau

Academic Editor

PLOS ONE

---

## [Editor Report · Acceptance letter]

2 Jun 2022

PONE-D-22-08912R1 

Impacts on tundra vegetation from heavy metal-enriched fugitive dust on National Park Service lands along the Red Dog Mine haul road, Alaska. 

Dear Dr. Neitlich:

I'm pleased to inform you that your manuscript has been deemed suitable for publication in PLOS ONE. Congratulations! Your manuscript is now with our production department. 

Kind regards, 

on behalf of

Dr. Tim A. Mousseau 

Academic Editor

PLOS ONE